



# Peat macropore networks – new insights into episodic and hotspot methane emission

Petri Kiuru[1], Marjo Palviainen[2], Tiia Grönholm[3], Maarit Raivonen[4], Lukas Kohl[5,6], Vincent Gauci[7,8], Iñaki Urzainki[1,9], and Ari Laurén[1]

[1]School of Forest Sciences, Faculty of Science and Forestry, University of Eastern Finland, P.O. Box 111, 80101 Joensuu, Finland
[2]Department of Forest Sciences, University of Helsinki, P.O. Box 27, 00014 Helsinki, Finland
[3]Finnish Meteorological Institute (FMI), Erik Palmenin aukio 1, 00560 Helsinki, Finland
[4]Institute for Atmospheric and Earth System Research (INAR)/Physics, Faculty of Science, University of Helsinki, P.O. Box 68, 00014 Helsinki, Finland
[5]Department of Agricultural Sciences, University of Helsinki, P.O. Box 56, 00014 Helsinki, Finland
[6]Institute for Atmospheric and Earth System Research (INAR)/Forest Sciences, Faculty of Agriculture and Forestry, University of Helsinki, P.O. Box 56, 00014 Helsinki, Finland
[7]Birmingham Institute of Forest Research (BIFoR), University of Birmingham, Edgbaston, Birmingham, B15 2TT, UK
[8]School of Geography, Earth and Environmental Sciences, University of Birmingham, Edgbaston, Birmingham, B15 2TT, UK
[9]Natural Resources Institute Finland (Luke), Latokartanonkaari 9, 00790 Helsinki, Finland

**Correspondence:** Petri Kiuru (petri.kiuru@uef.fi)

**Abstract.** Peatlands are important natural sources of atmospheric methane ($CH_4$) emissions. The emissions are strongly influenced by the diffusion of oxygen into the soil and of $CH_4$ from the soil to the atmosphere. This diffusion, in turn, is controlled by the structure of macropore networks. The characterization of peat pore structure and connectivity through complex network theory approaches can give insight into how the relationship between the microscale pore space properties and $CH_4$ emissions on a macroscopic scale is shaped. The formation of anaerobic pockets, which are local hotspots of $CH_4$ production in unsaturated peat, can also be conceptualized through a pore network approach. In this study, we extracted interconnecting macropore networks from three-dimensional X-ray micro-computed tomography (μCT) images of peat samples and evaluated local and global connectivity metrics for the networks. We also simulated the water retention characteristics of the peat samples using a pore network modeling approach and compared the simulation results with measured water retention characteristics. The results showed large differences in peat macropore structure and pore network connectivity between vertical soil layers. The macropore space was more connected and the flow paths through the peat matrix were less tortuous near the soil surface than at deeper depths. In addition, macroporosity, structural anisotropy, and average pore throat diameter decreased with depth. Narrower and more winding air-filled diffusion channels may reduce the rate of $CH_4$ transport as the distance from the peat layer to the soil–air interface increases. Hysteresis was found to affect the evolution of the volume of connected air-filled pore space in unsaturated peat. Thus, the formation of anaerobic pockets may occur in a smaller soil volume and methanogenesis may be slower when the peat is wetting compared to drying conditions. This hysteretic behavior should be taken into account in biogeochemical models to explain the hotspots and episodic spikes of $CH_4$ emissions. The network analysis also suggests that both local and global network connectivity metrics, such as the network average clustering coefficient and closeness centrality,





might serve as proxies for assessing the efficiency of $CH_4$ diffusion in air-filled pore networks. However, the applicability of

the network metrics was restricted to the high-porosity near-surface layer. The spatial extent and global continuity of the pore network and the spatial distribution of the pores may be reflected in different network metrics in contrasting ways.

## 1   Introduction

Peatlands are globally important modulators of hydrological and biogeochemical cycles (Holden, 2005; Limpens et al., 2008). They are major sources of carbon dioxide ($CO_2$) and methane ($CH_4$), thereby contributing to global warming (Frolking et al.,

2011; Abdalla et al., 2016; Leifeld et al., 2019). Peatland $CO_2$ and $CH_4$ emissions are influenced by management practices such as drainage and restoration, which affect the water table (WT) dynamics (Menberu et al., 2016; Evans et al., 2021) and therefore the air-filled porosity and the availability of oxygen ($O_2$) in peat (Waddington et al., 2015; McCarter et al., 2020). This, in turn, influences $CO_2$ and $CH_4$ emissions, as the oxidation of organic matter to $CO_2$ is stimulated under oxic conditions, whereas its reduction to $CH_4$ requires anoxic conditions.

Such anoxic conditions prevail below the WT but also in microniches – anaerobic pockets – in unsaturated soil above the WT (Silins and Rothwell, 1999; Deppe et al., 2010). These pockets form when the consumption of $O_2$ exceeds the transport, mainly diffusion of $O_2$. If readily degradable organic carbon is also available, the anaerobic pockets may become microscale hotspots of $CH_4$ production (Wachinger et al., 2000; Hagedorn and Bellamy, 2011). As the peat $CH_4$ concentration increases above the atmospheric concentration, $CH_4$ may diffuse through the aerobic peat layer into the atmosphere if it is not oxidized

by methanotrophic bacteria before reaching the peat surface (Whalen, 2005). Because the diffusion coefficient of $CH_4$ in air is 4 orders of magnitude higher than in water (Ball and Smith, 2001), the air-filled porosity of the unsaturated zone largely regulates the diffusional $CH_4$ transfer in peat and therefore determines how much $CH_4$ is oxidized before reaching the peat surface.

Water retention characteristic is a fundamental soil property that links soil structure to water and aeration dynamics, redox

conditions, and many accompanying biogeochemical processes (Bachmann and van der Ploeg, 2002; Lepilin et al., 2019). Soil water retention in the low suction range (0–10 kPa), which represents the filling and emptying of macropores (Perret et al., 1999), is strongly influenced by soil structure and pore size distribution (Hillel, 1998). Soil macropores are defined as pores having an effective diameter of the order of 100 µm or greater (Beven and Germann, 1982). Such low suction range conditions dominate in peatlands, where the WT is generally between 1 m and the soil surface (Sarkkola et al., 2010). This highlights the

importance of macropores in peat soil functions (Reddy and DeLaune, 2008; McCarter et al., 2020). Pore size distribution or the water retention characteristic alone does not provide information about the arrangement, connections, or topology of the pores. Macropores form a complex network, where the individual macropores can be open and connected, dead-ended, or isolated (Rezanezhad et al., 2016). The topology and structure of the network therefore regulates water, solute, and gas transport and, ultimately, biogeochemical processes in peat (see McCarter et al., 2020). Dead-ended and disconnected pores can block the gas

transfer between peat and the atmosphere and promote the formation of anaerobic pockets above the WT (Knorr et al., 2009;



Estop-Aragonés et al., 2012). Thus far, the role of anaerobic pockets in the $CH_4$ processes has been neglected in simulation models (e.g., Fan et al., 2014), mainly because the pore network is difficult to characterize experimentally.

Pore characteristics have been earlier described by pore size distribution derived from the water retention characteristic (Laine-Kaulio, 2011; Lepilin et al., 2019) or by tortuosity indices derived from air permeability measurements (Laurén, 1997), both methods assuming homogeneous and isotropic soil structure (Beckwith et al., 2003). However, the $CH_4$ production and transport cannot be fully understood without considering the three-dimensional (3D) pore network structures in peat. X-ray imagery and complex network theory provide a very promising yet unstudied approach for describing these structures and their effect on soil gas transport properties and mechanisms. X-ray micro-computed tomography (μCT) allows an explicit description of pore structure with resolution extending to micrometer scale (Perret et al., 1999; Blunt et al., 2013; Rezanezhad et al., 2016). Total porosity and pore size distributions can be determined directly from 3D images (Larsbo et al., 2014), and when the extracted pore space is represented as a 3D network of pores and pore throats, more detailed information about pore connections and topology can be obtained (Gostick, 2017). Methods of complex network theory are widely used for quantifying multi-scale connectivity and transport processes in real-world networks (Newman, 2003). Network concepts such as clustering, centrality, tortuosity, isolation, and path lengths can be used to characterize the macropore network from the anaerobic pocket formation viewpoint. This may be a key in explaining the observed hotspots and episodic spikes of $CH_4$ flux, which are particularly difficult to explain in the current $CH_4$ models (e.g., Xu et al., 2016).

The aims of this study were to introduce the μCT and complex network theory methods to analyze the characteristics of macropores and their networks in peat and to evaluate the characteristics from the viewpoint of gas exchange, $CH_4$ processes, and the formation of anaerobic pockets.

## 2 Materials and methods

### 2.1 Field sampling

Peat samples were collected from Lettosuo, which is a drained forested peatland site in southern Finland. The site was drained in 1969 with open ditch drains arranged in 40 m spacing. The study site belongs to the Integrated Carbon Observation System (ICOS), and it is located in Tammela, southern Finland (60° 38' N, 23° 57' E). The mean annual temperature and precipitation are 4.6 °C and 627 mm, respectively (Pirinen et al., 2012). The soil type is histosol dominated by *Carex* peat. The site was originally a mesotrophic fen classified as an herb-rich tall sedge birch-pine fen (Laine and Vasander, 1996). The dominating tree species, with a mean height of 20 m, are Scots pine (*Pinus sylvestris* L.) and Downy birch (*Betula pubenscens* Ehrh.) with an understory composed of Norway spruce (*Picea abies* Karst.). The stand volume is 230 $m^3$ $ha^{-1}$ with a density of 2200 stems $ha^{-1}$. The ground vegetation is composed of dwarf shrubs with a coverage of 4 % (*Vaccinium myrtillus* L., *V. vitis-idaea* L.) and herbs (coverage 10.6 %) such as *Dryopteris carthusiana* (Vill.) H.P. Fuchs and *Trientalis europaea* L. The moss layer is patchy and dominated by *Pleurozium schreberi* (Brid.) Mitt., *Dicranum majus* Turner, and *D. polysetum* Sw. In addition, *Sphagnum girgensohnii* Russow, *S. russowii* Warnst., and *S. angustifolium* (C.E.O. Jensen ex Russow) C.E.O. Jensen are present in moist patches. A detailed site description is available in Koskinen et al. (2014) and Bhuiyan et al. (2017).





Undisturbed peat samples were extracted into acrylic cylinders (diameter 50 mm, height 50 mm) using a sharp knife and
85  scissors. The samples were collected from seven randomly located pits and three different depths (0–5 cm, 20–25 cm and 40–
45 cm, hereinafter referred to as top, middle, and bottom layer, respectively). First, a 50–60 cm deep pit, with an undisturbed
vertical face, was dug, and then the profile depth was measured with a ruler. Vertically oriented peat samples were extracted
along the pit face paying attention to maintaining the undisturbed peat structure. The peat samples were located into plastic
bags and transported to the laboratory, where the water retention measurement was started immediately.

## 2.2  Measurement of water retention and air-filled porosity

The water retention characteristics of the cylindrical peat samples were measured in the laboratory using a pressure plate
apparatus (Hillel, 1998). The samples were saturated with water, weighed, and placed into the pressure plate apparatus. The
volumetric water content ($\theta$, m$^3$ m$^{-3}$) was then allowed to stabilize in the pressure plate apparatus under external pressures of
1, 3, 6, and 10 kPa for 1 week at each pressure level. These pressure levels are equivalent to soil matric potentials ($\Psi$) of $-1$,
$-3$, $-6$, and $-10$ kPa. The sample mass ($M_\Psi$, kg) was determined after each pressure level. At the end of the experiment, the
height and diameter of the sample were measured to determine the shrinkage, and then the samples were sent to µCT imaging
(see Sect. 2.3). After the imaging, the peat samples were dried in 105 °C for 72 h to obtain the dry mass ($M_s$) of the sample.

The bulk density ($\rho_b$, kg m$^{-3}$) of the sample was determined from the dry mass and the volume of the sample in the saturated
state ($V_{sat}$, m$^3$). The volumetric water content $\theta$ was calculated at each matric potential in relation to the saturated volume of
the sample as

$$\theta = \frac{(M_\Psi - M_s)/\rho_w}{V_{sat}} \tag{1}$$

where $\rho_w$ (kg m$^{-3}$) is water density. Total porosity $f$ (m$^3$ m$^{-3}$) was calculated as

$$f = 1 - \frac{\rho_b}{\rho_s} \tag{2}$$

where the particle density $\rho_s$ was assumed to have the value of 1500 kg m$^{-3}$ (Redding and Devito, 2006). Air-filled porosity
$f_a$ was determined at each matric potential as the difference of total porosity and the respective volumetric water content. In
addition, it was assumed that no entrapped air was present in the samples and the air-filled porosity was zero in the saturated
state.

Water retention properties were also characterized using the van Genuchten model (van Genuchten, 1980)

$$\theta = \theta_r + \frac{(\theta_s - \theta_r)}{[1 + (\alpha|\Psi|)^n]^{1/(1-n)}} \tag{3}$$

where $\theta_r$ is the residual water content, $\theta_s$ is the saturated water content, and $\alpha$ and $n$ are empirical fitting parameters. The fitting
was performed for the average values of $\theta$ of the samples from each depth by applying the Levenberg–Marquardt algorithm
in SciPy (Virtanen et al., 2020). Because the applied pressure range was rather narrow, the fitting procedure was simplified
by setting the saturated water content to equal the total porosity and setting the residual water content to zero as suggested by
Weiss et al. (1998).



## 2.3 Three-dimensional µCT imaging

In short, X-ray micro-computed tomography involves taking two-dimensional X-ray photographs of an object from multiple angles and then using a filtered back-projection algorithm to reconstruct the 3D volume of the sample. With the method it is possible to get noninvasively information about the internal structure of the sample and to visualize and quantify the sample. For example in soil studies, it is possible to determine the pore characteristics (porosity and pore size distribution), grain size distribution, and moisture distribution inside the sample (Taina et al., 2008; Helliwell et al., 2013).

Soil samples (see Sect. 2.1) were scanned in the micro-CT laboratory in the University of Helsinki with the GE Phoenix Nanotom system. The final voxel (cubic 3D image element) size after reconstruction was 50 µm, and the data were stored in an unsigned 16-bit integer representation. The size of the resulting 3D images was 1142 by 1142 by 1152 voxels. Some darkening was observed in many of the images near the top and bottom of the cylindrical samples, which may have resulted from defects in µCT image reconstruction.

## 2.4 Image processing

In the µCT image preprocessing stage, the 16-bit 3D grayscale images were converted to 3D binary images that represent the void and solid volumes of the samples. This was done using the Python image processing packages SciPy ndimage (Virtanen et al., 2020) and scikit-image (van der Walt et al., 2014) and the image analysis toolkit PoreSpy (Gostick et al., 2019).

First, the 3D grayscale images were straightened through rotation and cropped to a size of 1000 by 1000 by 1000 voxels. A cylindrical peat volume (height 1000 voxels, diameter 1000 voxels) excluding the acrylic cylinder was separated using PoreSpy. Before noise filtering and binary segmentation, the 16-bit images were linearly mapped to an unsigned 8-bit representation. The 16-bit to 8-bit mapping interval was selected for each image by visual inspection of the grayscale histogram so that the long, shallow tails of the intensity distribution, which were mainly generated by noise, were removed. The resulting 8-bit grayscale images were then filtered for noise reduction with a 3D median filter with a radius of 2 voxels (Fig. 1a). The intensity contrast between air-filled regions and the regions containing water or organic matter in the images was often rather low (Fig. 1b). Furthermore, not all the grayscale histograms were bimodal so that the intensity values for void and solid regions could not be readily distinguished from the intensity distributions. The segmentation of images into void and solid volumes was performed using the widely utilized Otsu's global thresholding method (Otsu, 1979) (Fig. 1c). Finally, isolated solid volumes were removed from the binary images using a method for finding disconnected voxels in PoreSpy.

## 2.5 Pore network extraction

Pore networks were extracted from the binarized $1000^3$-voxel images using a marked-based watershed segmentation method (Gostick, 2017) available in PoreSpy (Figs. 1d and 2). The extraction method has been designed to have good performance also for materials with a high porosity. It generates the topology of the pore network by dividing the void space into individual pore regions and determining the locations of pore throats, that is, the two-dimensional interfaces between adjacent pores, and the connections between the pores. The method facilitates the subsequent determination of pore network geometry, which





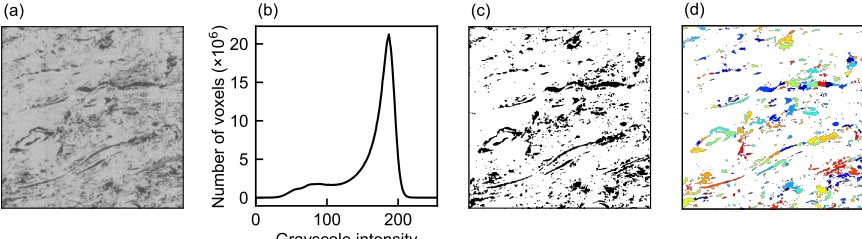

**Figure 1.** (a) Vertical slice of a $600^3$-voxel section of the noise-filtered X-ray micro-computed tomography image of one of the 40–45 cm peat samples. (b) Histogram of the grayscale intensities of the corresponding noise-filtered cylindrical 8-bit image with a height and diameter of 1000 voxels. (c) Vertical slice of the binary image resulted from solid–void segmentation. White region denotes solid material and black regions denote void space. (d) Division of void space into individual pore regions.

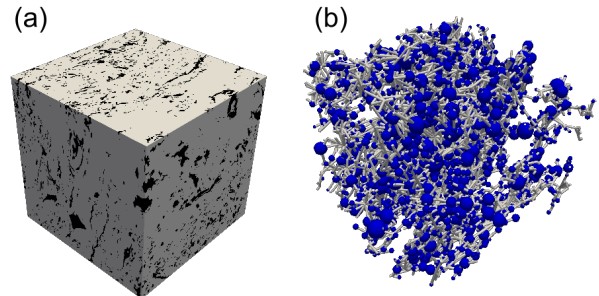

**Figure 2.** (a) $600^3$-voxel (27 cm$^3$) cubic central region of the binary image generated from the X-ray micro-computed tomography image of one of the peat samples from the depth of 40–45 cm. Gray region denotes solid material and black regions denote void space. (b) Largest cluster of interconnected pores, or a pore network, extracted from the cubic image. The presentation of the network follows stick-and-ball geometry: pores are depicted as blue spheres, and throats are gray straight cylinders between connected pores.

includes, for example, pore volumes, pore-to-pore distances, and throat diameters. Feature resolution is generally about twice the voxel size in µCT imaging (Stock, 2008), implying that the size of the smallest distinguishable feature was 100 µm.

## 2.6 Pore geometry

The size of a pore was characterized by its volume, which was determined by counting the number of voxels in an individual pore region. The diameter of a pore was defined as the diameter of the largest sphere that fits inside the pore region. Similarly, the throat diameter was defined as the diameter of the largest circle that fits inside the throat region. Further, the equivalent pore diameter, defined as the diameter of a sphere with the same volume as the pore, was used in pore size classification in the study. The distance between adjacent pores was determined as the sum of the distances between the centroids of each pore and

the centroid of the throat between them.





## 2.7 Image and network analyses

### 2.7.1 Image and network porosity

The porosities of the binary images were calculated using the cylindrical sections of the $1000^3$-voxel binary images. In the calculations, it was assumed that the sample surfaces were in level with the ends of the acrylic cylinder in the initial state.

The image porosities were determined as the ratios of the number of void voxels to the number of total voxels in the cropped image section covering the inner space of the cylinder. It was thus assumed that the shrinkage of the peat sample resulted in the displacement of inner void space into the space between the sample and the walls of the cylinder.

The extracted pore system can be partitioned into clusters of interconnected pores and a set of isolated individual pores. The largest of these clusters is defined as the pore network. The total pore system, including also smaller pore clusters and

isolated pores, is thereafter referred to as total pore space. Network porosity is defined as the ratio of the sum of the volumes of individual pores in the network to the total volume of the applied sample domain. The volume of the total pore space is slightly smaller than the volume of the void space of the corresponding section of the binary image because the network extraction algorithm tends to discard some of the smallest isolated pores especially adjacent to the borders of the image domain.

### 2.7.2 Water retention simulation

Water retention simulation was performed employing the algorithm for drainage percolation in the open-source pore network modeling package OpenPNM (Gostick et al., 2016). In the algorithm, a pore network is initially filled with a defending fluid (water). An invading fluid (air) enters the network through inlets located in a specified boundary region of the network and gradually replaces the defending fluid under increasing external pressure as in a porosimetry experiment. Fluid invasion is access-limited, which means that the invading fluid can only enter the throats that are directly connected to the inlet. The

pressure $P$ needed to force the invading fluid to penetrate a throat and enter the adjoining pore is determined by the Washburn equation as

$$P = -\frac{4\gamma\cos\beta}{D} \tag{4}$$

where $\gamma$ is the interfacial tension between the invading and defending phases (0.72 mN m$^{-1}$ for an air–water interface), $\beta$ is the contact angle of the invading fluid, and $D$ is the diameter of the throat. The contact angle was assumed to be 180° in

the simulations, which gave the maximum limit of the throat entry pressure. The entry pressures corresponding to different contact angles can be readily estimated because the Washburn equation is linear with respect to $\cos(\beta)$. The air-filled porosity at each external pressure can be calculated as the volume fraction of air-filled pores. Water imbibition was simulated with the percolation algorithm by using site percolation, in which fluid invasion pressure is controlled by pore diameters instead of throat diameters.

The network domain size used in the water retention simulations had to be as close to the total sample size as possible so that comparison with the measured retention curves would be reasonable. Thus, only 100 voxels, representing 5 mm slices, were excluded from the top and bottom of the sample images in order to exclude the roughness of the sample surfaces and





the influence of decreased grayscale intensity near the horizontal image boundaries on solid–void classification. The network domain was determined to be 40 mm in height, and it included the whole cylindrical region in the horizontal direction. Because

of a slight vertical shrinkage in some of the top layer samples, the height of the network domain was decreased to 30 or 35 mm as needed. The resulted image was then divided into four regions of similar shape with horizontal dimensions of 500 by 500 voxels. A separate pore network was extracted for each of these image regions with PoreSpy. Water retention simulations for the four subnetworks were performed using the same pressure steps in each simulation. The combined air-filled porosities at each pressure step were then calculated to represent the water retention characteristic in the total cylindrical network domain

with a height of 800 voxels and a diameter of 1000 voxels. The maximum external pressure applied in the simulations, 2.88 kPa, was determined by the minimum throat diameter, which was 100 μm.

### 2.7.3 Network metrics

In order to exclude the effects of shrinkage, a centered cubic subregion with a side length of 600 voxels (30 mm), hereafter referred to as a subsample, was selected from each sample image for the analysis of pore size distribution and connectivity.

Hence, the results characterized better the actual inner structure of a sample. The largest connected pore cluster, or the pore network, extracted from the cubic subregion was used in the network connectivity analyses. In order to determine the air-filled volume fraction of a pore network at different external pressures, we also performed drainage and imbibition percolation simulations for the cubic-domain networks.

We used the network analysis package NetworkX (Hagberg et al., 2008) for the estimation of network connectivity metrics.

The pore *coordination number* (the degree of a node in graph theory) gives the number of connections to an individual pore, or in other words, the number of throats emanating from a pore. The local *clustering coefficient* of a pore A is defined as the probability that two pores that are connected to A are also connected to each other. The network average clustering coefficient, the average of all local clustering coefficients (Watts and Strogatz, 1998), can be considered as the probability that two adjacent pores of a random pore are connected to each other. The *closeness centrality* of a pore is defined as the reciprocal of the average

of the shortest path lengths from the pore to every other pore in the network (Freeman, 1978). The calculated pore-to-pore distances were used as the edge weights in the calculation of the path lengths. The closeness centrality of the pore network was calculated as the average value of the closeness centralities of all the pores. A high network closeness centrality indicates that the overall global connectivity of the pore network is relatively high (van der Linden et al., 2019).

Geometrical tortuosity and betweenness centrality are, respectively, network measures related to the transport properties of

a spatial network in a certain direction and as a whole. To characterize properties related to the connectivity through the peat pore network between the opposite surfaces of the network domain, artificial boundary pores were added to the pore network to represent the interfaces at the surfaces of the samples using PoreSpy.

The *geometrical tortuosity* or path length tortuosity of a pore network is defined as the average value of the ratio of the lengths of the shortest paths between each pair of pores located at the opposite boundaries of the network domain to the

straight-line distance between the opposite boundary planes (Lindquist et al., 1996; Clennell, 1997). In the calculation of geometrical tortuosity, the shortest paths between boundary pores were determined using Dijkstra's algorithm. Geometrical





tortuosity was determined separately for the vertical and for the horizontal direction. Both perpendicular horizontal directions were included in the calculation of horizontal geometrical tortuosity.

The *betweenness centrality* of a pore A is generally defined as the ratio of the number of shortest paths between all the pairs of pores in the network that traverse A to the total number of the pairs of pores that do not include A (Freeman, 1977). In this study, we also defined and quantified the *top–bottom betweenness centrality* by including only the shortest paths that connected the top boundary pores with the bottom boundary pores. As with closeness centrality, the betweenness centrality of a pore network was determined as the average value over the pores. A high network betweenness centrality suggests that the shortest paths through a network tend to be governed by a relatively small number of different routes (van der Linden et al., 2019). In order to save calculation time, every tenth pore was used in the estimation of the network average closeness centrality, betweenness centrality, and geometrical tortuosity in networks with more than 8000 pores.

## 2.8 Statistics

In order to determine if the water retention characteristics, pore sizes, or calculated network metrics differed between depths, we applied a one-way analysis of variance (ANOVA) followed by Tukey's pairwise multiple comparison test. If residual normality or variance homogeneity could not be assumed, a nonparametric Kruskal–Wallis test followed by Dunn's pairwise multiple comparison test or Welch's ANOVA followed by Games–Howell pairwise multiple comparison test, respectively, were applied instead. A paired sample *t*-test was applied to analyze the difference between vertical and horizontal geometrical tortuosity. The statistical analyses were conducted with the statistical function module in SciPy and the Python packages statsmodels (Seabold and Perktold, 2010), scikit_posthocs (Terpilowski, 2019), and hypothetical (Schlegel, 2020).

## 3 Results

### 3.1 Air-filled porosity

The void fractions of the μCT images corresponded well to the measured air-filled porosities of the samples (Fig. 3). A slight discrepancy was found in the middle layer samples. The measured vertical shrinkage of the samples at $-10\,\mathrm{kPa}$ matric potential decreased with depth, being on average 6.3 % in the top layer samples, 3.7 % in the middle layer samples, and 2.3 % in the bottom layer samples. The total pore space volumes within the cylindrical network domains were 0.4–2.4 %, 1.4–9.7 %, and 1.9–6.6 % smaller than the void space volumes in the corresponding image sections of the top, middle, and bottom layer samples, respectively. The network volumes, on the other hand, were 0.2–3.0 %, 6.3–35.1 %, and 6.6–27.8 % smaller than the total pore space volumes, as some of the void space consisting of small, isolated void regions was omitted during network extraction.

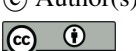



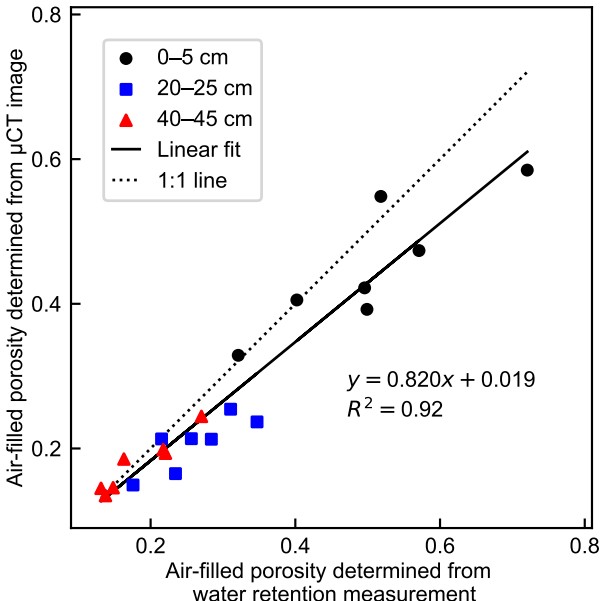

**Figure 3.** Measurement-derived (determined by subtracting the volumetric water content from total porosity) and image-derived (determined as the fraction of void voxels in binarized µCT image) air-filled porosities of the cylindrical peat samples from different depths at $-10$ kPa matric potential.

## 3.2 Water retention and air invasion dynamics

The total porosity of the peat samples differed significantly between sampling depths (Welch's ANOVA, $F(2, 10.56) = 7.83$, $p = 0.008$), but the differences were generally rather small (Table 1). The between-sample variation in porosity was highest in the top layer. Also, the water retention characteristics differed significantly between depths (ANOVA or Kruskal–Wallis test, $p < 0.05$, Table 1). Water content in all studied matric potentials was clearly lowest in the top layer and highest in the deepest studied peat layer.

The air-filled porosities derived from measurements and pore network simulations were rather coherent at different matric potentials (Fig. 4). In the pore network simulations, the external pressure range extended only to about 3 kPa, which corresponds to the minimum throat diameter of 100 µm detected in the µCT imaging. In the samples with no notable shrinkage in any direction, such as top layer samples 3, 6, and 7 and the middle layer sample 7, the percolation simulation matched the measured air-filled porosity values well at both $-1$ kPa and $-3$ kPa matric potentials. The hysteresis effect is clearly seen in the drainage–imbibition simulations (Fig. 5). The air-filled porosity at a certain matric potential was higher during water imbibition than during drainage.





**Table 1.** Means and standard deviations for the bulk density ($\rho_s$, kg m$^{-3}$), total porosity ($f$), and volumetric water content ($\theta$) of the peat samples at different matric potentials and the fitted van Genuchten water retention parameters $\alpha$ (cm$^{-1}$) and $n$. Different letters indicate significant difference between depths ($p < 0.05$).

| | $\rho_s$ | $f$ | $\theta(-1\ \mathrm{kPa})$ | $\theta(-3\ \mathrm{kPa})$ | $\theta(-6\ \mathrm{kPa})$ | $\theta(-10\ \mathrm{kPa})$ | $\alpha$ | $n$ |
|---|---|---|---|---|---|---|---|---|
| 0–5 cm | 140±29ab | 0.907±0.019ab | 0.604±0.121a | 0.551±0.116a | 0.479±0.112a | 0.403±0.100a | 0.43 | 1.21 |
| 20–25 cm | 152±14a | 0.898±0.009a | 0.779±0.040ab | 0.755±0.049ab | 0.696±0.058b | 0.638±0.053b | 0.22 | 1.10 |
| 40–45 cm | 125±9b | 0.917±0.006b | 0.888±0.016b | 0.860±0.025b | 0.786±0.039b | 0.733±0.044b | 0.017 | 1.26 |
| p | 0.008 | 0.008 | < 0.001 | < 0.001 | < 0.001 | < 0.001 | | |
| test$^a$ | Welch | Welch | Kruskal | Kruskal | ANOVA | ANOVA | | |

$^a$ ANOVA: F-test and Tukey's test; Welch: Welch's F-test and Games–Howell test; Kruskal: Kruskal–Wallis test and Dunn's test.

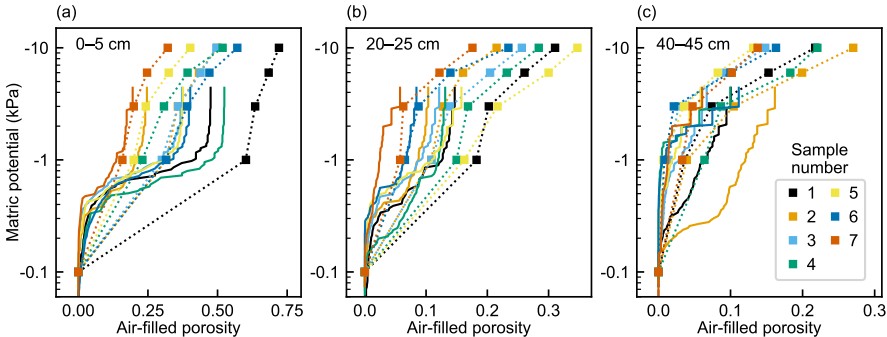

**Figure 4.** Measurement-derived (square markers with dotted connecting lines) air-filled porosities of peat samples from (a) 0–5 cm, (b) 20–25 cm, and (c) 40–45 cm depths at different matric potentials and the corresponding simulated air invasion curves (solid lines). Air-filled porosity was determined by subtracting the measured volumetric water content from total porosity.

### 3.3 Pore and throat size distributions

The pore sizes of the connected pore networks obtained from the $600^3$-voxel subregions of the peat sample images ranged from 265   $4 \times 10^{-4}$ mm$^3$ to 75 mm$^3$, which corresponds to an equivalent diameter range of 0.09 mm to 5.2 mm (Fig. 6a). The median pore size of the top layer networks (median equivalent diameter $0.66 \pm 0.06$ mm) was significantly larger than that of the middle ($0.58 \pm 0.01$ mm) and bottom ($0.59 \pm 0.02$ mm) layer networks (Kruskal–Wallis test, $H(2) = 9.67$, $p = 0.008$). Also, the between-sample variation in pore size distribution was substantially higher in the top layer (the medians of the equivalent diameters 0.579–0.765 mm) than in the deeper layers (medians 0.573–0.589 mm and 0.565–0.617 mm in the middle and 270   bottom layers, respectively). The maximum pore sizes were generally highest in the top layer networks, but large pores also existed in some of the bottom layer networks (Fig. 6c-e). A larger total volume of a pore network was related to a larger average pore size in the top layer networks but not in the deeper layer networks.



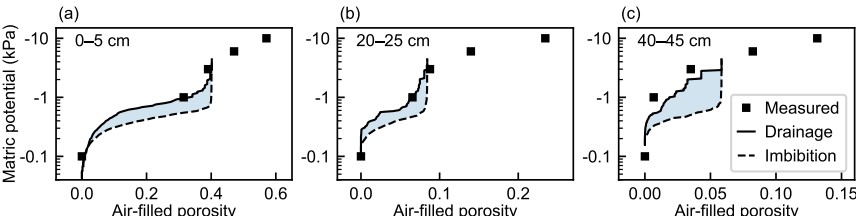

**Figure 5.** Simulated drainage and imbibition curves for selected peat samples from (a) 0–5 cm, (b) 20–25 cm and (c) 40–45 cm depths and the corresponding measured air-filled porosity values.

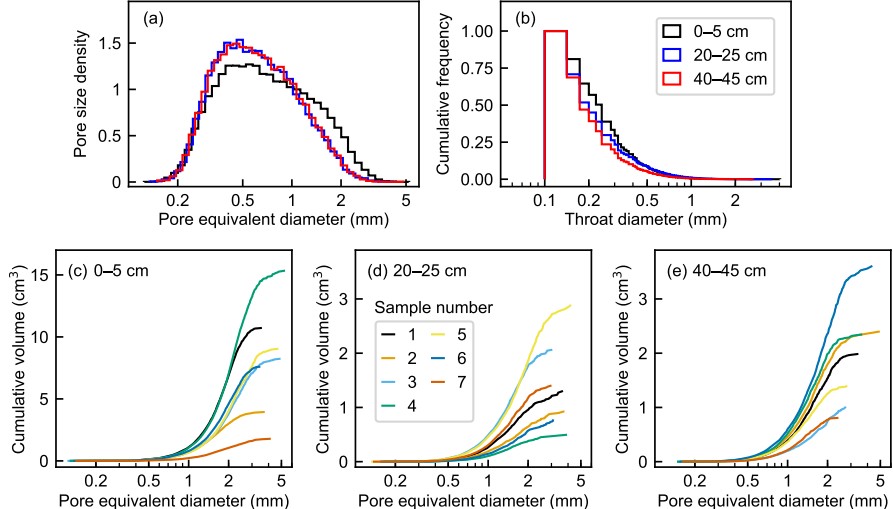

**Figure 6.** Pore size characteristics of the $600^3$-voxel domain pore networks from the sampling depths of 0–5 cm, 20–25 cm and 40–45 cm. (a) Combined histograms of the probability density functions of pore size. (b) Combined cumulative relative frequency histograms of throat diameters. (c–e) Cumulative volume distribution of each pore network in relation to pore equivalent diameter.

The throats with the smallest detectable diameter (100 μm) were the most abundant at all depths, and the fraction of wider throats decreased with depth (Fig 6b). However, the difference between the mean throat diameters at different depths was marginally nonsignificant (ANOVA, $F(2, 18) = 2.82$, $p = 0.09$).

### 3.4 Network porosity and connectivity metrics

The porosity of the pore network and all the pore network metrics differed significantly among depths (ANOVA or Kruskal–Wallis test, $p < 0.05$, Fig. 7). Network porosity and connectivity were clearly highest in the top layer. The connected pore networks did not extend over the whole cubic network domain in most of the middle and bottom layer subsamples and in one of the top layer subsamples, which decreased the obtained network porosity. All the network metrics of the top layer differed



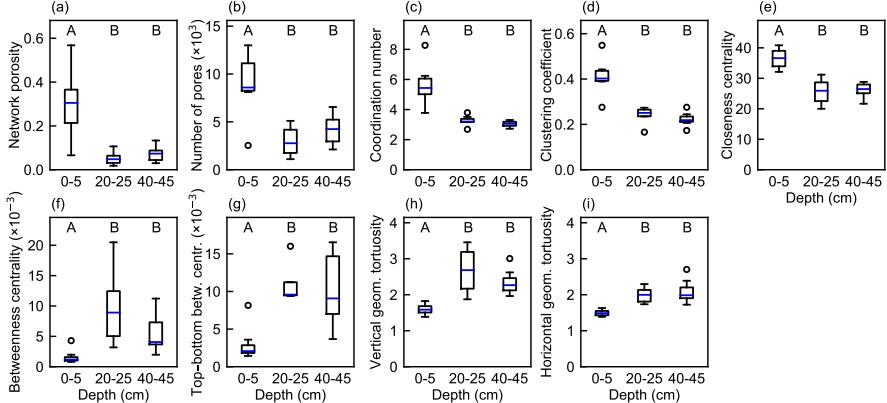

**Figure 7.** Boxplots of properties and metrics of pore networks generated from peat samples from different depths ($n = 4$ for vertical geometrical tortuosity and top–bottom betweenness centrality at 20–25 cm, otherwise $n = 7$). (a) Porosity of the pore networks; (b) number of pores in the networks; (c) network average coordination number; (d) network average clustering coefficient; (e) network average closeness centrality; (f) network average betweenness centrality; (g) average betweenness centrality for paths between top and bottom boundary pores; (h) average geometrical tortuosity of the network in the vertical direction; (i) average geometrical tortuosity of the network in the horizontal direction. Boxes indicate the interquartile range, whiskers extend at most 1.5 times the interquartile range from the first and from the third quartile, and the blue line shows the median. Letters not shared across depths represent significantly different means via the pairwise comparison tests ($p < 0.05$). Kruskal–Wallis test and Dunn's post hoc test were used for coordination number and clustering coefficient, otherwise ANOVA and Tukey's post hoc test were applied.

**Table 2.** Correlation coefficients between vertical geometrical tortuosity and different network connectivity metrics for pore networks at different depths. The coefficients are shown in italics if the relationship is not significant ($p > 0.05$).

| | Coordination number | Clustering coefficient | Closeness centrality | Betweenness centrality | $n$ |
|---|---|---|---|---|---|
| 0–5 cm | −0.89 | −0.91 | −0.69 | *0.42* | 7 |
| 20–25 cm | *0.79* | *0.52* | *0.12* | −0.97 | 4 |
| 40–45 cm | *0.07* | *0.31* | *−0.26* | *−0.10* | 7 |

significantly from those of the deeper layers ($p < 0.05$), whereas no significant difference was observed between the middle and bottom layer subsamples.

The vertical geometrical tortuosity was significantly higher than the horizontal geometrical tortuosity in the top layer networks (mean difference 0.10; two-tailed paired sample $t$-test, $t(6) = 3.67$, $p = 0.01$), but no significant difference was found at the deeper layers. Furthermore, the variation of geometrical tortuosity in the vertical direction was slightly higher than in the horizontal direction (Fig. 7h,i). In the top layer, vertical geometrical tortuosity decreased with increasing network average coordination number, clustering coefficient, and closeness centrality and increased with increasing network average betweenness centrality, but the behavior was very different and inconsistent in deeper layers (Table 2).



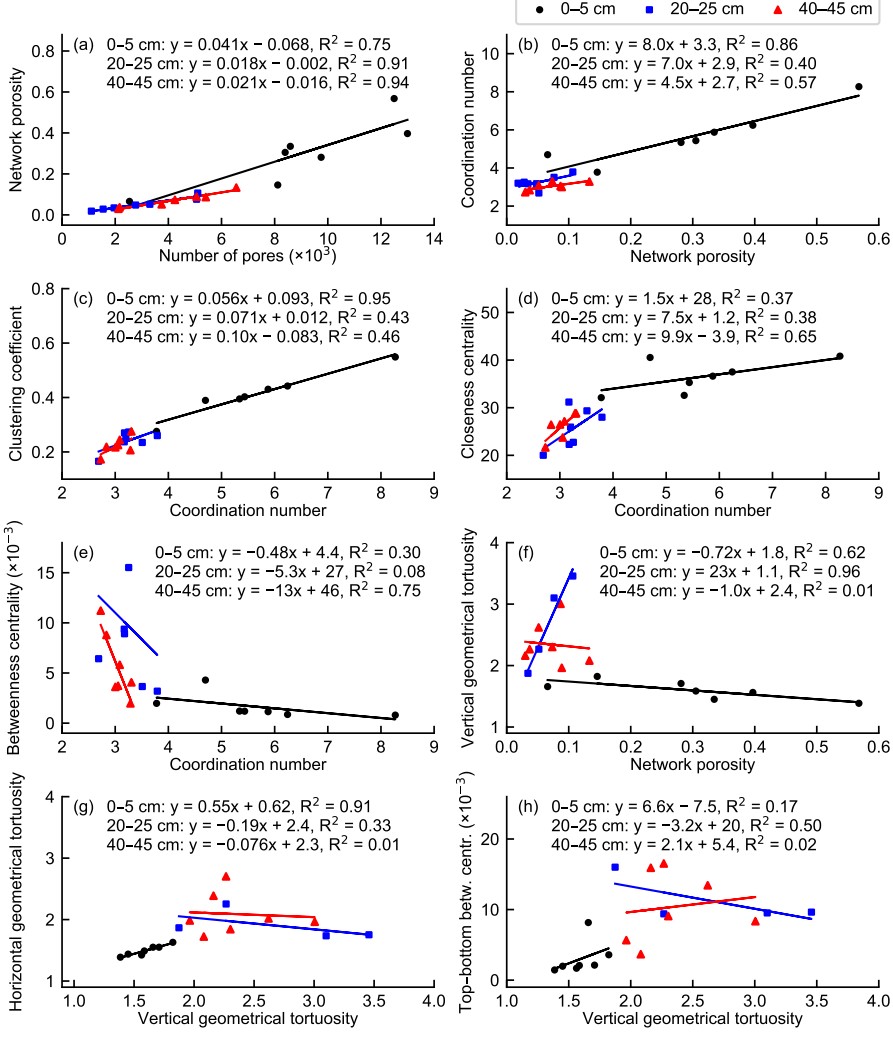

**Figure 8.** Linear relationships among selected pore network properties and average pore network connectivity metrics.

The total volume of the connected pore space increased as a function of the number of pores in the network, but the variation

was higher in the top layer subsamples than in the middle and bottom layer subsamples (Fig. 8a). Pore sizes tended to increase with increasing porosity in the top layer subsamples, whereas this connection was not found for the deeper layers. The network average coordination number, characterizing local network connectivity, increased with increasing porosity in the top layer subsamples, in which the connected pore space was rather evenly distributed in the network domain (Fig. 8b). In the middle and bottom layer networks, the correlation between the coordination number and porosity was slightly weaker. The network

average clustering coefficient also increased with increasing coordination number in all the studied layers (Fig. 8c).



The network average closeness centrality increased with increasing coordination number in the top layer except for the sub-sample with smaller network dimensions and largely also in deeper layers (Fig 8d). By contrast, high local network connectivity indicated a low betweenness centrality especially in the top and bottom layers (Fig. 8e). The connected pore cluster extended from the top to the bottom face of the network domain only in four out of the seven middle layer subsamples. Vertical geomet-

rical tortuosity and top–bottom betweenness centrality could thus not be determined for the remaining three networks.A higher network porosity indicated a lower vertical tortuosity in the top layer networks (Fig. 8f). By contrast, the middle and bottom layer networks showed high variation in vertical tortuosity in relation to network porosity. Horizontal and vertical geometrical tortuosity were strongly correlated in the top layer networks, in which the spatial distributions of the pores within the networks were rather uniform (Fig. 8g). In the deeper layers, horizontal and vertical tortuosity were unrelated. Also, vertical geometrical

tortuosity and top–bottom betweenness centrality were rather well correlated in the top layer but not in deeper layers (Fig. 8h).

### 3.5    Air-filled volume fraction

Figure 9 describes the volume fraction of the connected, air-filled pore network of the total pore space at different external pressures. Values less than 1 indicate that a part of the pore space has been isolated from the surrounding volume and $O_2$ supply has ceased. Eventually, the isolation can lead to the formation of an anaerobic pocket. The volume fraction of the

connected network was calculated for both imbibition (wetting) and drainage (drying). In imbibition, half of the network remained connected at 0.5 kPa in all layers, whereas in the drainage simulation this required 1 kPa in the top layer and 3 kPa in the middle and bottom layers. The layers behaved rather similarly in imbibition, but in drainage there were clear differences between the layers.

The fraction of the total pore network volume of the total pore space volume, which corresponds to the volume fraction at an

external pressure of 3 kPa (Fig. 9h), was largest in the top layer subsamples, extending from 93.6 % to 99.9 %. In the deeper subsamples, 7.4 to 51.9 % of the total pore space was disconnected from the largest pore cluster. Thus, a significant fraction of macropore space was inactive in the drainage and imbibition simulations in the deeper layer subsamples, which indicates an even larger pore volume available for anaerobic pocket formation.

### 4    Discussion

### 4.1    Evaluation of image and network analysis in macropore characterization

The combination of X-ray tomography, image analysis, and network analysis provides detailed information on pore structures, connections, and topology that cannot be obtained through traditional laboratory methods. These properties determine gas exchange in peat and are therefore essential in regulating biological activity. One of the aims of our study was to evaluate the applicability of image analysis and network extraction methods to peat structure characterization. Overall, the image-derived

porosities and the simulated water retention characteristics corresponded rather well with the laboratory measurements given the limitations imposed by the applied imaging resolution. The smallest pore diameter detectable in the µCT imaging was

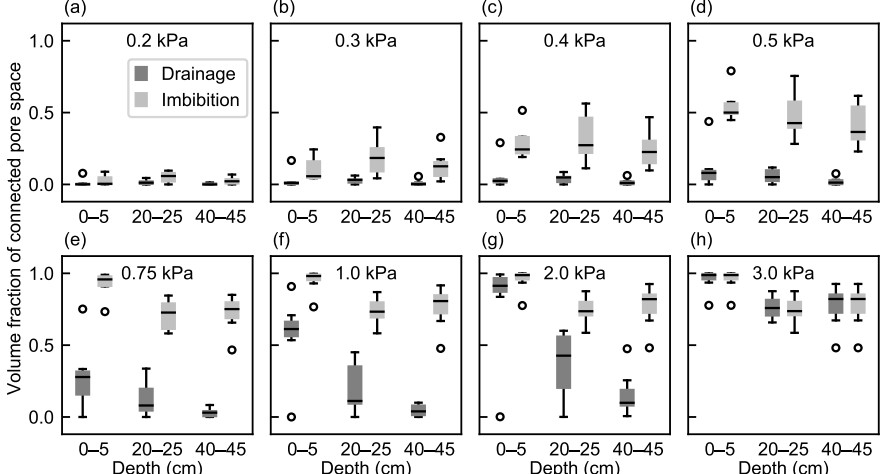

**Figure 9.** Volume fractions of air-filled pore networks of total peat macropore space at external pressures of (a) 0.2 kPa, (b) 0.3 kPa, (c) 0.4 kPa, (d) 0.5 kPa, (e) 0.75 kPa, (f) 1.0 kPa, (g) 2.0 kPa, and (h) 3.0 kPa in drainage and imbibition simulations ($n = 7$ in 0–5 cm and 40–45 cm, $n = 5$ in drainage and $n = 6$ in imbibition in 20–25 cm). Boxes indicate the interquartile range, whiskers extend at most 1.5 times the interquartile range from the first and from the third quartile, and the blue line shows the median.

100 µm, corresponding to a matric suction of approximately 3 kPa. This resolution is sufficient for accurately describing diffusional gas transport in peat soils, where matric suction typically remains low and pores smaller than 100 µm are generally water-filled. By comparison, macropores have been found to dominate water and solute transport in peat (McCarter et al., 2020), and a similar resolution is also considered adequate for simulating hydrological transport processes in water-saturated peat (Gharedaghloo et al., 2018).

Good performance of image segmentation is a crucial prerequisite for a successful application of µCT image analysis and subsequent quantitative analysis tools (Iassonov et al., 2009). The binary segmentation stage succeeded fairly well in our study. The sample void fractions obtained through µCT image analysis were in good agreement with the respective air-filled porosities derived from laboratory measurements (Fig. 3). The slightly lower values obtained from µCT image analysis likely resulted from the limited feature resolution of the images. The matric potential of the soil during µCT imaging ($-10$ kPa) corresponds to the pressure needed to penetrate a pore with a diameter of approximately 30 µm, whereas the minimum dimension of a detectable void in the images was 100 µm. The number of air-filled pores with diameters between 30 and 100 µm may have been largest in the middle layer samples, which showed the largest offset between the two measurement methods. By contrast, the global solid–void classification method seems to have overestimated the void volume in some of the images of the top layer samples. Further, the darkening of the µCT images due to image reconstruction defects near the top and bottom of the sample increased the void fraction slightly in all binary images because the darkened areas were falsely classified as void space at the binary segmentation stage.





The pore network extraction stage introduced another level of complication in macropore volume characterization. The

porosities of the networks used in the water retention simulations were further diminished with respect to the corresponding image porosities, especially in deeper layers. The main reason for this was that a considerable fraction of pore space was disconnected from the active pore network, especially in some of the middle and bottom layer samples. This was because narrower void connections may have not been detectable in the images, which may have resulted in some pore clusters or individual pores becoming isolated from the main connected cluster. In addition, some of the reduction of pore space connectivity and of

the total, combined volume of connected pore space resulted from the division of the total network domain into four discrete regions.

The shrinkage behavior of peat further obscured the determination of network porosity. In the networks with a centered cubic domain, the average porosity was slightly lower in the middle layer subsamples than in the bottom layer subsamples (Fig. 7a), while the opposite was the case in the larger-domain cylindrical networks (Fig. 4). In addition to the spatial heterogeneity of

air-filled porosity within the samples, the difference can be explained by the variation of horizontal shrinkage between depths. Higher shrinkage in the middle layer samples decreased the air-filled porosity of the samples, which was reflected in the air-filled porosity of the cubic network domain, which did not include the void region between the shrunk sample and the cylinder walls.

Peat shrinkage also affected the results of the water retention simulations. Ideally, the simulated air-filled porosity at the

maximum external pressure should have been close to the corresponding measured air-filled porosity at $-3$ kPa matric potential, which corresponds to the minimum detectable pore throat dimension in the images (Fig. 4). However, because of the limited image resolution, the constructed pore geometry may not totally represent the actual void space geometry under the conditions of $-3$ kPa or higher matric potentials because the shrinking of the samples may have resulted in a decrease in void space. Thus, a fraction of the pore throats that were air-filled a $-3$ kPa matric potential may have shrunk so that they were

not detectable in the µCT images constructed at $-10$ kPa matric potential. This may have generated disconnected pore space and decreased the total volume of the extracted pore network. Also, shrinkage may have decreased the dimensions of the pore space so that a higher external pressure was needed for air invasion in the simulations. Conversely, the horizontal shrinkage of some of the samples created continuous void space near the cylinder wall at $-10$ kPa matric potential, and thus the extracted pore network contained pore space that had presumably not been present at higher matric potential conditions.

**4.2 Networks, anaerobic pockets, and methane processes**

Emissions of $CH_4$ from northern peatlands are characterized by a large spatiotemporal variation (Abdalla et al., 2016; Rinne et al., 2018). The $CH_4$ efflux usually occurs episodically and in hotspots (Lai, 2009), and the supply of $O_2$ and gas diffusion conditions in the soil profile are the main abiotic factors affecting $CH_4$ emissions (Xu et al., 2016). We argue that the structure, topology, and behavior of peat macropore networks above the WT can be a good candidate for explaining the spatiotemporal

variation of peatland $CH_4$ emissions. The following conceptualization describes the pore network, anaerobic pocket formation, and $CH_4$ production and transport: When the pore network is internally connected and open to the atmosphere, the supply of $O_2$ to the soil is adequate and facilitates aerobic decomposition. When some of the pores become blocked by water, the $O_2$





supply to the soil gradually decreases, as the number of air-conducting pores decreases and the air flow paths become longer and more tortuous. With increasing water content, part of the pore network becomes isolated and the $O_2$ supply is prevented.

Next, microbial activity consumes the trapped $O_2$, and the microbial metabolism changes to alternative electron acceptors. In the final stage, the production of $CH_4$ onsets. An anaerobic pocket has formed. When soil water content decreases again, the network becomes connected, and the $CH_4$ trapped in the pocket is released and can be detected as a burst of $CH_4$ emission at the soil surface.

The existence of isolated anaerobic macropore clusters above the WT may also enable $CH_4$ production in newly aerated

layers during a drying period (Knorr et al., 2008; Estop-Aragonés and Blodau, 2012). Large macropores are drained and exposed to $O_2$ as the WT declines, but pore clusters with narrower throats adjacent to the air-filled regions remain anoxic if diffusional $O_2$ supply to these water-filled pores is inadequate. If the isolated macropores are formed close to the peat surface, the time span that a substance needs to diffuse to the atmosphere is relatively short and, consequently, there is a limited exposure to oxidation in the aerated layer. Thus, methanogenesis in isolated near-surface macropores may significantly increase

the atmospheric emission of $CH_4$ (Estop-Aragonés et al., 2013). In addition, anaerobic pore clusters may be the regions where anaerobic microbes may survive above the WT and facilitate the onset of $CH_4$ production when the WT rises again (Kettunen et al., 1999).

Our drainage and imbibition simulations indicated that hysteresis may affect the formation and destruction of anaerobic pockets and the temporal $CH_4$ dynamics in the unsaturated layer. Figure 9 describes conditions under which the anaerobic

pockets are likely to occur. When the peat is drying, the macropore network remains largely water-filled until a matric potential of $-2$ to $-3$ kPa, which under hydrostatic equilibrium corresponds to a distance of 20–30 cm from the WT. This promotes a high abundance of anaerobic pockets in the unsaturated layer close to the WT. The thickness of this layer is greater at deeper depths, where the average pore throat dimensions are smaller. In a hectare scale, this suggests that 2000–3000 $m^3$ of unsaturated peat is potentially active in $CH_4$ production. Under wetting conditions, the macropore network remains largely air-filled until

the matric potential is higher than $-1$ kPa or the distance to the WT is less than 10 cm. This means that the thickness of the layer where anaerobic pockets can be formed when the WT is rising is less than 10 cm. The large difference between the thickness of the pocketing layer under drying and wetting conditions is likely to cause a short-term mismatch between peatland $CH_4$ emissions and WT observations. For example, the hysteresis effect may contribute to the lagged response of $CH_4$ flux to a rising WT that has been observed in peatland ecosystems (Kettunen et al., 1996; Goodrich et al., 2015).

The average pore volumes and pore throat diameters were smaller at deeper depths than in the near-surface peat (Fig. 6). The reduction of pore space volume and pore dimensions with depth, which is due to increasing degree of decomposition of peat and higher compression by overlying matter (Rezanezhad et al., 2016), is a typical feature of peat soil (Rezanezhad et al., 2010; Gharedaghloo et al., 2018). Thus, the macropore network is drained at lower matric potentials in deeper peat than near the soil surface. In our samples, virtually all macropores were connected at $-3$ kPa matric potential in the top layer, whereas

roughly one-quarter of the macropore volume was still isolated in deeper layers (Fig. 9). As a result, the width of the layer favorable for anaerobic pocket formation is greater in deeper, more decomposed peat. However, the lower macroporosity and





smaller average volume of pores may reduce the total volume of anaerobic pockets and limit the capacity for methanogenesis
in the pocketing layer at deeper depths.

In our study, the peat samples originated from a drained forested peatland site. The total porosity range of the studied peat
layers was well in line with the values and the vertical variation reported for the near-surface layers of drained peatlands in
the literature (Päivänen, 1973; Minkkinen and Laine, 1998). However, the peat soil of a drained peatland differs in physical
characteristics and pore size distribution from that of an undisturbed peatland (Liu et al., 2020). Pore deformation due to peat
shrinkage induced by drainage is partly irreversible (Price, 2003), and the pressure of a growing tree stand may further compact
the peat layer and increase its bulk density in deeper layers (Minkkinen and Laine, 1998). Also, the saturated peat layer below
the WT may be further compressed due to the weight of overlying peat that has become drier and lost the support by buoyancy
(Hooijer et al., 2012; Sloan et al., 2019). These processes decrease the macroporosity of the peat profile in drained peatlands in
comparison to undisturbed peatlands (Liu et al., 2020). Therefore, conditions for methanogenesis may be even more favorable
in undisturbed peatlands because of a potentially larger volume of macropore space available for anaerobic pocket formation.

### 4.3   Network connectivity metrics related to gas transport

Van der Linden et al. (2016) highlights the need for multiscale measures of pore space topology and connectivity and the rela-
tionship of these descriptors to macroscopic transport processes in a porous medium, such as peat. We utilized several kinds of
network metrics for this purpose. The pore coordination number and the clustering coefficient were used to characterize the lo-
cal connectivity of a pore network, whereas closeness centrality and betweenness centrality were used to describe connectivity
in the network scale. Furthermore, top–bottom betweenness centrality and geometrical tortuosity characterize the positioning,
shape, and length of gas transport routes through the porous medium in a certain direction, thus describing the efficiency of gas
transfer through the medium. It is essential to determine how the local and global network connectivity measures are related to
the network gas transfer capability estimated by vertical tortuosity. According to our results, high average local pore connec-
tivity is not always reflected as high global connectivity or gas transfer capability of the pore space. The distribution and spatial
coverage of the connected pore space within the porous medium was found to regulate the applicability and comparability of
the local and global measures and also their relationship to porosity and tortuosity.

Generally, a higher network porosity implied higher local pore connectivity characterized by the coordination number and
the clustering coefficient (Fig. 8b,c). However, if the connected pore space was concentrated in a smaller region in the network
domain, which was the case in most of the deeper layer subsamples, local connectivity was rather high even though the
porosity of the network was relatively low. Likewise, the relationship between local and global connectivity measures was
largely dictated by the shape and topology of the network. In general, higher local network connectivity was reflected as a
higher average closeness centrality (Fig. 8d), which means that the path lengths between pores shortened when more alternative
paths were available. The average betweenness centrality mainly decreased with increasing average coordination number at all
depths, which indicates that the shortest routes between pairs of pores were spread out more widely when the number of pores
and their connectivity was higher (Fig. 8e). However, some variation existed especially at deeper layers, where the porosity
was lower and the spatial extent of the network was typically smaller.





The location of a pore largely determines its centrality in a spatial network because the probability of a pore being part of the shortest route between two other pores is highest near the centroid of the network (Barthélemy, 2011). High abundance of dead-end pores or pores located near the extremities of the network decreases the average betweenness centrality because these pores do not belong to the shortest paths between other regions. In addition, the closeness centrality of a pore (the reciprocal 450 of the average of the shortest path lengths from a pore to every other pore) is related to the dimensions of a network within its domain. If a network covers only a fraction of its domain or if it contains dense local clusters with narrow pore channels between them, the average closeness centrality may become relatively high even if the average coordination number is rather low.

The ratio between the two local connectivity measures may reflect the structure of a pore network. A high ratio of the network 455 average clustering coefficient to the network average coordination number indicates that the number of three-pore loops is relatively high and may also imply that even larger clusters of interconnected nearby pores are abundant in the network. By contrast, a low ratio may result from a high abundance of long and narrow pore conduits in the network, which may diminish the amount of alternative transport routes within the network and lead to a rapid suppression of gas transport if pores become clogged.

The geometrical tortuosity of a pore network in a certain direction is strongly dependent on the geometry of the connected cluster within the network domain and on the localization of the boundary pores. If the boundary pores are spread evenly on the opposite surfaces of the domain and the internal pores are also uniformly distributed in the network domain, the geometrical tortuosity is relatively low even if the network porosity is low. By contrast, if the number of boundary pores is low and they are located in a small area or if there are constrictions in the pore space, the average length of the shortest paths between the 465 boundary pores may be high even if the total number of the pores is high and the network is otherwise well connected.

Our results also showed a correspondence between the two network measures related to the transport properties of a network, geometrical tortuosity and top–bottom betweenness centrality, in the top layer networks, most of which had a spatially rather even pore distribution (Fig. 8h). If the average shortest path length between the top and bottom boundary pores is low (the vertical geometrical tortuosity is low) and the boundary pores are located evenly on the top and bottom surfaces of the domain, 470 the shortest paths between different pairs of boundary pores are also located more evenly throughout the network (the average top–bottom betweenness centrality is low). Because of a smaller number and more localized spatial distribution of boundary pores and more variable network topology and geometrical structure, there was evidently no correspondence between vertical geometrical tortuosity and top–bottom betweenness centrality in the deeper layers.

A low geometrical tortuosity of a pore network suggests that gas transfer through the network is efficient in a specified 475 direction. In the top layer subsamples with a higher network porosity and a stronger correlation between vertical tortuosity and network porosity (Fig. 8f), a lower vertical geometrical tortuosity was strongly related to network metrics values that indicated higher network connectivity (Table 2). By contrast, such a correlation was not found for any local or global connectivity metrics in other layers. Thus, the applicability of network connectivity metrics to the description of network transport properties seems to be highly sensitive to the spatial distribution of the network within its domain. The average closeness centrality of a pore 480 network has been found to be a proxy for the efficiency of fluid flow in porous media (van der Linden et al., 2019). The





simulation of diffusion through a peat pore network and the comparison of different network metrics with effective diffusivity could give further insight on the relation of network theory measures to macroscale $CH_4$ transfer processes in peat. In addition to network average connectivity metrics, analysis of the distributions of different connectivity measures of individual pores within a network could further illustrate the properties of peat pore structure.

The structural anisotropy of peat, characterized by the difference between vertical and horizontal geometrical tortuosity, can be used to estimate the efficiency of gas diffusion through peat in different directions. In this study, the pore structure was slightly anisotropic in the top layer, but no anisotropy was found in deeper layers. Thus, structural anisotropy decreased with lower network porosity and pore connectivity. However, the geometrical tortuosity was highly variable between samples in deeper layers because of the very heterogeneous spatial distribution of the pores in the middle and bottom layer networks. The

variation of anisotropy between depths suggests that chains of pores existed between horizontally orientated, less degraded plant residues, giving rise to a low horizontal geometrical tortuosity. In deeper levels with more degraded and compacted peat, the horizontally oriented pore chains had fractured and collapsed, and the structural anisotropy had diminished. These findings are in line with Kruse et al. (2008) and Liu et al. (2016) who found that the anisotropy of the hydraulic conductivity of fen peat decreased with increasing degradation. According to our results, the orientation of diffusion paths in pore networks does

not promote or restrain gas transfer towards the atmosphere in deeper, more degraded peat layers. In less degraded peat near the surface, the impact of the orientation of plant litter on the primary diffusion direction may be more pronounced. However, higher pore connectivity and porosity in the less degraded peat may outweigh the hindering effect of structural anisotropy on the rate of gas diffusion from the peat to the atmosphere.

## 5   Conclusions

The network analysis of the peat pore system enabled by μCT imaging and a network representation of peat macropores provides new insights into the impact of pore structure and pore space connectivity on conditions regulating $CH_4$ production and transfer in peat. As the WT is generally close to the surface and low suction conditions prevail in peatlands, small pores remain continuously water-filled and diffusional gas transfer occurs in the air-filled macropore network above the WT. We argue that the complex pore structure and the vertical variation in the pore characteristics of peat may promote the formation

of anaerobic pockets above the WT during fluctuations of soil water content. When the WT finally declines and soil water content decreases, $CH_4$ produced in these pockets can be released rapidly via air-filled macropores. In addition, hysteresis was found to regulate the thickness of the zone favorable for anaerobic pocket formation. Under the same WT, the pocketing can be distinctively different depending on whether the peat is wetting or drying. This may provide an explanation for the observed hotspots and episodic spikes of $CH_4$ emissions in peatlands. Network analysis demonstrated fundamental differences in peat

pore structure and macropore characteristics between topsoil and deeper soil layers. Pore space was more connected and routes through the peat matrix were less tortuous in the top layer than in deeper layers. Decreasing pore connectivity with depth was accompanied by a lower number of macropores, smaller macropore volumes, and narrower pore throats. This may indicate that the rate of $CH_4$ diffusion in the air-filled pore space is reduced in deeper peat layers.

The results also suggest that local and global network connectivity metrics might be used to estimate the efficiency of diffusional $CH_4$ transfer in the air-filled pore space of peat. However, we highlight that connectivity metrics should be evaluated with caution because not only pore connectivity but also the extent and spatial distribution of connected air-filled pore space regulate the $CH_4$ transfer capabilities. Most importantly, a pore network representation of peat macropore structure enables the application of pore network modeling, which is a useful method for the pore-scale description of $CH_4$ production and transfer processes in peat and for the investigation of relations between peat pore structure and $CH_4$ dynamics.

*Code and data availability.* The data that support the findings of this study are available in Github [https://github.com/pjkiuru/macropore_networks]. The μCT image, binary image, and pore network data are available from the corresponding authors upon reasonable request. The image processing and simulation parts of this study used the publicly available packages PoreSpy and OpenPNM. The Python scripts used in the calculations and the simulation output are available in Github [https://github.com/pjkiuru/macropore_networks].

*Author contributions.* AL, TG, and IU developed the idea and designed the study. AL and MP collected the samples and performed the water retention measurements. TG and MR organized the μCT imaging and 3D reconstruction. PK processed the images, designed and conducted the computations and simulations, and analyzed the data. PK and LK performed the statistical analysis. PK, AL, and MP wrote the manuscript with significant contributions from TG, VG, and IU. All other authors provided edits and comments on the paper. AL and MR are responsible for the funding acquisition.

*Competing interests.* The authors declare that they have no conflict of interest.

*Acknowledgements.* This research has been supported by the Academy of Finland (grant nos. 325168 and 325169). LK holds a Marie Skłodowska-Curie actions fellowship under the European Commission's Horizon 2020 program (grant no. 843511). AL was supported by the funding from the Academy of Finland to strengthen university research profiles in Finland for the years 2017–2021 (funding decision 311925). MR acknowledges SRC at the Academy of Finland (SOMPA, no. 312932) and EU-H2020 (VERIFY, no. 776810). This work used services of the Helsinki University X-Ray Micro-CT Laboratory, funded also by Helsinki Institute of Life Science (HiLIFE) under the HAIP platform.





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
