# Peer review of "Peat macropore networks – new insights into episodic and hotspot methane emission"

_Biogeosciences, 2021_

## Author Comment (AC2)

bg-2021-259
**Author responses to comments of Referee #2**

We would like to thank the referee for the effort and time he put in to comment on our manuscript. We are grateful for his careful and considered comments and will make every attempt to fully address these comments in the revised manuscript.
In the following list, the points raised by the referee are written in bold characters, whereas our responses are shown in regular characters.

**The manuscript is a carefully detailed and well described study on pore network analyses of peat soils with a depth differentiated view. The aim is to use results of pore network analyses to help explain methane production potential in these environments. The level of language is very high, the text flow is very good, and it was a great pleasure to review. Although I am not that familiar with micro CT, image analyses and pore network simulations, the results are feasible, while not super surprising, as would be expected from the literature. Still, the manuscript is novel in its approach and comparisons.**

**I was left a little bewildered, what had happened to the discussion of methane (L371-383 gives some general statements on the importance and general conceptual discussion), in particular, since it was not explicitly modelled nor were the results contrasted against methane flux measurements from the field.**

**My recommendation is that, now with some distance, the authors re-assess the paper and whether or not the self set aims are fully fullfilled. Perhaps it might be a good idea to focus a little more on the strong technical part of the manuscript and limit the study a little more to the descriptive nature, and then discuss the methane production and diffusion from this vantage points in a more speculative manner. After all, the emerging methane emissions are mostly explained by a conceptual model.**

The aim of our work is to study the capability of complex network theory methods and metrics to characterize the physical structure of peat and its pore space and to qualitatively assess the obtained metrics and correlations between them in relation to gas transfer processes in peat. The study also shows the physical existence and evolution of isolated air-filled pockets and illustrates their capability to control the production and emission of gases, and methane in particular, from peat. A more thorough comparison between the pore network modelling method and experimental methods related to gas diffusion efficiency in peat will be the subject of further studies. We will revise and reformulate the study aims in the last paragraph of the Introduction. We will also restructure the Discussion and Conclusions so that the conceptual nature of the assessment of methane-related processes will be indicated more clearly.

Major concern

**What does this study have to do with methane emissions? This remained unclear to me. Either, the scope of the study should be a more quantitative**

The study is part of a larger project that aims to develop tools for studying and modeling methane emissions from peat. The study aims to create a conceptual basis for the description of processes related to methane generation and its atmospheric emissions. The methods presented and used in this study to assess the structure and connectivity of peat pore space can later be applied to describe the processes related to gas transfer and biogeochemical processes in peat. Specifically, we aim to qualitatively evaluate the potential to use the pore network characteristics revealed by µCT imaging and the complex network theory metrics calculated for these networks for assessing gas transfer and processes related to organic matter degradation in peat. Methane generation generally requires anoxic conditions, and studying the development and evolution of regions prone to anoxia in peat illustrates the capabilities of peat for methane generation. We will revise the aims of the study and restructure the Discussion and Conclusions so that the conceptual nature of the assessment of methane-related processes will be indicated more clearly.

Minor concerns

L5 How can the formation of anaerobic pockets be conceptualized in a pore network approach? This is left unaddressed.

The formation and evolution of pore regions isolated from the atmosphere during peat wetting can be assessed and illustrated using pore network modeling. Performing a porosimetry simulation in a pore network gives a description of the gradual change of the number of air-filled pores and the development of isolated pore clusters. Therefore, a pore network modeling approach offers a way to conceptualize the physical basis of the formation of anaerobic pockets. This is illustrated in the modeling results in Sect. 3.5. We will clarify the issue and reformulate the sentence in the abstract.

L42-43 Here peat specific citations should be made (e.g. Hayward and Clymo (1982), Weber et al., 2017), after all, the suggest a pore size distribution.

We will add the suggested references.

L53 like previous comments.

We will add the suggested references.

L 90 ff section 2.2: was the VGM model fitted to the averages of the replicates from each depth, respectively? Just a minor information to add.

Yes. We will rephrase the description.

L94 Sentence not needed, but not harmful, either.

We will keep the sentence for clarity.

L97 in -> at

We will correct the preposition.

L105 This is a rather bold statement. If samples were not saturated under co2 environment of vacuum, my impression is this is not actually correct. Please do not use this assumption.

We did not use the assumption in the calculation of the air-filled porosity of the samples. The mass of the saturated sample was not used in the determination of the total porosity, as the total porosity was calculated using the measured dry mass and an estimated, constant particle density (Eq. 2). Therefore, the assumption that air-filled porosity was zero in the saturated state did not affect the air-filled porosity calculations at other matric potentials. The assumption was only reflected in the fitting of the van Genuchten model, where we state that the value of the saturated water content parameter was assumed to equal the total porosity that we had estimated. In addition, the assumption was used in Fig. 4 as the measured air-filled porosity value at -0.1 kPa matric potential, but this value is not essential in the analysis of the water retention properties presented in the figure. The assumption of full saturation is also consistent with the fact that the pore network was, by definition, filled with water at the initial state of the porosimetry simulation.

Because the assumption is used only for illustrative purposes in Fig. 4, we will remove the statement from this section and state the choice and give reasons for it in the caption of Fig. 4.

L118 quantify the sample: What is meant by this? Please specify.

We meant that the structural properties, such as pore space volume and pore size distributions, can be assessed quantitatively. We will clarify the sentence.

L122 is resolution meant, here?

In tomography, voxel size and (spatial) resolution are different concepts. Voxel size is the spatial dimension represented by one image element. Spatial resolution is related to the size of the smallest distinguishable feature in the image, and it is roughly twice the voxel size in microtomography.

L122 if this is the resolution, then the real micropores <10micron are not resolved. Thus, much of the area where anaerobia may continue to exist is not covered. Please address this limitation in the discussion. This, alongside the problem of dissecting organic from water.

Our study is focused on peatlands or drained peatlands, where the water table depth is less than 1 m. It is therefore assumed that pores less than 30 µm in diameter are essentially water-filled under these conditions. Further, the conceptual discussion on anaerobic pocket formation is not focused on the specific networks analyzed in this study. The argumentation can be generalized to pore networks with any pore dimensions.

Separating water from organic matter was not an issue in this study. The purpose of image segmentation was to identify the air-filled volume of the sample images. The problem concerned dissecting air from other substances than air. The word 'solid' stands for both water and organic

material in the text. The low intensity difference between regions containing gaseous material and regions containing non-gaseous material made it difficult to determine an unambiguous boundary between air and other material. We will clarify the definition of solid in Sect 2.4. and address the role of the low intensity contrast as an error source in the Discussion.

L191 The resulting image

We will correct the sentence.

L198ff you state you exclude the effect of shrinkage (I think you should neglect it in this analyses), but I am not convinced this method does what you say it does. Perhaps elucidate a little more.

We meant that horizontal shrinkage had generated empty space between the sample and the cylinder walls and near the top and bottom of the cylinder. If, for example, we had chosen a cylindrical region with a diameter of 50 mm (cylinder diameter) and a height of 25 mm, it would have included continuous void space generated by shrinkage near the cylinder walls. The selection of a study domain in the middle of the sample ensured that this extra void space was not included in the network domain. We will clarify this point in the description.

L213 relatively? to what?

We meant 'fairly'. We will correct the sentence.

Figure 3: Are the results shown for 3 samples, only, are there replicate samples included in the data? Potentially, including uncertainties in x might help explain the deviation from the 1:1 line (Table 1). Also, I expect that since the air filled porosity = 0 at saturation assumption is not warranted, the data points would be shifted to the left in x. This could be explored a little more. Also, what would the intercept in Figure 3 represent?

All the air-filled porosities plotted in the image represent the values at -10 kPa matric potential, and all the seven replicate samples are therefore included in the scatter plot in Fig. 3. We will emphasize this in the figure caption. The µCT imaging of the samples was performed at -10 kPa, and the images correspond to the porosity conditions at -10 kPa. The standard deviations presented in Table 1 are not the uncertainties of individual measurements but show the variation between the replicates at each depth. The assumption that the air-filled porosity was zero at saturation was not used in the determination of the air-filled porosity at -10 kPa. Total porosity was estimated from measured dry bulk density and estimated particle density (Eq. 2 in the manuscript). Air-filled porosity was obtained from volumetric water content (Eq. 1) and total porosity.

L256: "rather coherent": what does this mean, specifically? Contrary to this statement, in Figure 4 I see quite some systematic deviation.

Our description of the correspondence between the simulations and the measurements is indeed rather optimistic. The air-filled porosities derived from measurements and pore network simulations

were rather coherent at different matric potentials for some of the samples, but considerable variation existed between the samples at all depths. We will reformulate the sentence and extend the statement.

Figure 4: exhibits bimodality in the retention curve (i.e. some macroporisity close to saturation) due to sharper drop around 1 cm pressure head. This is visible at the top, and not at the bottom as expected from pedogenesis, again Weber et al. 2017.

In this study, the pore size distributions were obtained and analyzed primarily through the pore network approach, and the larger fraction of macroporosity in the top layer is seen in Fig. 6a. The decrease in individual pore volumes with depth is discussed in Sect. 4.2. We will elaborate this issue and include discussion about the retention curves in the Discussion.

Figure 4: It appears that a normalization was done onto one air filled porosity. correct? If so, please specify.

Absolute, not relative, air-filled porosities are presented in Fig. 4.

Figure 4: why are these simulations not carried out until a pressure head of -100cm? (Same in Figure 5).

As we state in Sect.3.2, the external pressure range used in the simulations extended only to about 3 kPa, which corresponds to the minimum throat diameter of 100 μm detected in the μCT imaging. The pore networks were therefore emptied of water at a pressure head of -30 $cmH_2O$, and nothing would have happened in the simulations as a result of a further increase in pressure.

L 290: This pattern can, again, be observed in Weber et al. (2017), but also in other sources in the literature. I suggest adding citations, here.

Disappearance of the largest pores and a smaller spatial variability of the pore size distribution in greater depths observed by Weber et al. (2017) can be inferred from Fig. 8a that shows that the variation between the total pore network volume and the number of pores was largest in the top layer and decreased with depth. However, Fig. 8a only indicates that the spatial variation of average pore volume is largest in the top layer. We prefer not to use citations in the Results section. We return to the changes in pore space properties with depth in the Discussion (Sect. 4.2). We will further elaborate this issue and refer to the literature in the Discussion.

L384-385 repetition of introduction

We would prefer to keep this sentence for continuity and as a transition to the subject of the paragraph.

L393: a bit of a leap of faith or perhaps a rather general statement: the simulation provide support for the conceptual understanding to be ok. I think these hard earned results should perhaps be discussed a little more in acknowledgement to this (although I detect absolutely no oversell, here).

We showed that the evolution of isolated air-filled pore space with respect to changes in matric potential was considerably different in drying and wetting. The air-filled pores with no connection to the atmosphere are prone to anoxia. This may give reason to suppose that also the development of anaerobic pockets differs in wetting and drying. The susceptibility of unsaturated peat to anoxia and methane generation is discussed in more length in the previous paragraphs of the section. We will elucidate and justify the argument further in the text.

L425: I see the potential to discuss the obtained pore space characterizing numbers in the light of other peoples results re: methane transport, or any other gas transport in particular. If this does not exist, I suggest to scope this section a little more carefully.

We use geometrical tortuosity as a proxy for diffusion efficiency and assess the applicability of network theory measures to characterize the transport properties of a network by comparing them with geometrical tortuosity. Geometrical tortuosity is, by definition, a direct proxy for network transport efficiency as it illustrates the tortuosity and length of flow paths through a network. Centrality measures (closeness centrality, betweenness centrality) are more advanced metrics adapted from complex network theory. Using geometrical tortuosity as a proxy, we could estimate qualitatively how well the centrality measures and other network theory metrics are able to describe the transport properties of a network. To our knowledge, there are no previous experimental studies on the use of complex network theory metrics for gas transfer in natural porous media. We will reformulate the title and the scope of Sect. 4.3 and emphasize the role of geometric tortuosity as a proxy in performance analysis.

L448 mark you, many of the dead end pores might not have been resolved due to the micro CT technique. Perhaps this should be contextualize, too.

The paragraph concerns the structure of a pore network and the characteristics and possible weaknesses of centrality measures in general, with no reference to a specific realization of a network or the relation of a network to a real porous object it represents. We think that a reference to the specific network is not relevant in the context of this paragraph.

L486: How can a relative quantitative measure (anisotropy) determine the efficiency (a qualitative) absolute measure of gas diffusion.

We did not mean the quantitative performance measure named efficiency but the capability of gas diffusion in different directions. We will change the wording.

References

Hayward, P. M. and Clymo, R. S.: Profiles of Water Content and Pore Size in Sphagnum and Peat, and their Relation to Peat Bog Ecology, Proceedings of the Royal Society B: Biological Sciences, 215, 299–325, https://doi.org/10.1098/rspb.1982.0044, 1982.

Weber, T. K. D., Iden, S. C., and Durner, W.: A pore-size classification for peat bogs derived from unsaturated hydraulic properties, Hydrol. Earth Syst. Sci., 21, 6185–6200, https://doi.org/10.5194/hess-21-6185-2017, 2017.

---

## Author Response (AR1)

**bg-2021-259 Author responses to Referee comments**

We would like to thank the referees for the effort and time they put in to comment on our manuscript. We are grateful for their careful and considered comments and will make every attempt to fully address these comments in the revised manuscript.

In the following list, the referee comments are written in bold characters, whereas our responses are shown in regular characters. The line numbers in the referee comments correspond to line numbers in the original version of the manuscript, whereas the line numbers in our responses refer to the revised manuscript.

**Referee #1**

This manuscript presents a thorough imaging and modeling study on pore networks in peat samples collected at different depths from a drained peatland, and uses this analysis to discuss implications of peat porosity and pore network characteristics on gas transport and methane production in peatlands. The manuscript is well written and easy to follow. The methods are explained well, although more details of error analyses and impacts of errors on the results, as well as some reconsideration of the statistical analyses would be needed (see specific comments below). My main concern about the presentation is that the results seem mostly intuitively obvious. It makes sense that porosity increases in peat towards the surface just because of how increasing pressure would influence and deform peat structures the deeper one goes in the soil layer. Maybe this has not been shown with the methods used in this study before, but the manuscript fails to communicate the state of the art in this kind of studies. This, combined with that one of the main results, the hysteresis behavior, seems to be just a model result with no verification based on the experiments, and the superficiality of the discussion on the connections between the porosity and network parameters and their impact on methane emissions and gas transport makes it hard to understand the importance and impact of the study. Was the goal to demonstrate what the used techniques can do? Or was it more to showcase a way to understand CH4 emissions?

In mineral soil, porosity decreases with depth because of compaction caused by overlying matter. However, in addition to physical factors such as pressure, air-filled porosity at a specific suction in peat is governed by the degree of peat decomposition. There was no decreasing trend in total porosity with depth (Table 1), which is consistent with other studies on drained peatlands (Minkkinen and Laine 1998). With increasing peat decomposition, larger pores fracture and collapse and the volume fraction of smaller pores increases. Therefore, macroporosity (air-filled porosity under low suction conditions) decreases with depth. The degree of decomposition generally increases with depth, which decreases the macroporosity in deeper peat (Rezanezhad et al. 2016). We have discussed this fact in Discussion (Sect. 4.3 in the revised manuscript), but we have now included it briefly also in the Introduction (L52–53).

We use more advanced methodology for the investigation of peat porosity and structure than most of the previous studies on the issue. The description of the pore space obtained through pore network analysis is more detailed in our study than in studies that have focused on the bulk properties of peat. The fact that our results are in line with an intuitive impression and the results of previous studies confirms the applicability of our method. A more detailed knowledge on pore space connectivity obtained by our method is needed for a better description of transport dynamics in peat.

Hysteresis, the dependence of the state of soil water on the process leading up to it, is a well-known phenomenon in soil science (Hillel 1998). It is observed as the difference between the volumetric water content of soil at a specific suction measured in sorption (wetting of soil when the water table is rising) and in desorption (drying of soil when the water table is falling), The experimental determination of the sorption curve is much more difficult than the determination of the desorption curve, and therefore it is seldom done. We show that pore network modelling can be successfully used to reproduce the hysteretic behavior of water retention and characterize the hysteresis effect in peat. Air-filled porosity is directly related to volumetric water content, and thus the evolution of connected pore space is a direct consequence of the evolution of water content. Therefore, hysteresis of air-filled porosity can be induced from hysteresis of water content. We have included a brief description of hysteresis in the Introduction (L19).

The aim of our work is to study the capability of complex network theory methods and metrics to characterize the physical structure of peat and its pore space and to qualitatively assess the obtained metrics and correlations between them in relation to gas transfer processes in peat. The study also shows the physical existence of anaerobic pockets and illustrates their capability to control the production and emission of gases from peat. A more thorough comparison between the pore network modelling method and experimental methods related to gas diffusion efficiency in peat will be the subject of further studies. We have reformulated and clarified the aims of the study in the last paragraph of the Introduction (L75–79). We have also restructured the Abstract, Discussion, and Conclusions so that the conceptual nature of the assessment of methane-related processes will be indicated more clearly.

This same unclarity in focus is apparent from the figures. Figure 3 is clearly a model vs. measurement comparison, but it is discussed in the results as a figure presenting characteristics of the peat samples. Figure 4, on the other hand, is presented in a format that emphasizes differences in the peat sample properties, but is discussed in the text as a model-measurement comparison. To improve the presentation, I suggest that the authors clearly define the purpose of the study, and then build the text and presentation of data in the figures to support that goal. For example, to support the statements about Fig 1, the changes in porosity between depths could be more effectively presented as a box or bar plot with appropriate error bars and statistical analyses. And for fig 4, the model-measurement comparison would be presented more clearly by an x-y –plot.

The discussion on Fig. 3 in Sect. 3.1 is intended to be focused on the comparison of the two methods of determination of the air-filled porosity of the samples, that is, on the assessment of the performance of image analysis. The procedures of the determination of air-filled porosity through water retention measurements and  $\mu$ CT imaging are different. The water retention measurements yield the volumetric water content of soil, and the air-filled porosity is then deduced using the estimated value of total porosity. By contrast, air-filled porosity is the quantity that can directly be determined from the binarized  $\mu$ CT images. Because the volumetric water content is the actual

measured quantity and the air-filled porosity is a derived quantity, it is more reasonable to analyze the volumetric water content statistically. The changes in volumetric water content between depths are presented in Table 1 with statistical analysis.

The latter half of the discussion in Sect. 3.1 is not related to Fig. 3 but considers the effect of the pore network generation process on the volume of the pore network in comparison to the void volume of the corresponding image section. The network volumes were smaller than the image section void volumes, which affected the results of the water retention simulation.

The graphs in Fig. 4 show the evolution of air-filled volume as a function of matric potential and the approximate shapes of the porosity–soil water potential curves. The graphs illustrate the variation between samples as well as the differences between simulation and measurements visually and explicitly. Fig. 4 is intended for qualitative analysis of the performance of the water retention simulations and of the pore network generation process behind them for each individual sample. Because the network domain used in the water retention simulations was smaller than the volume of the samples, it is not reasonable to accurately compare the measured and simulated values presented in Fig. 4 in more detail. Also, the measurement-based air-filled porosity values in Fig. 4 are not precise but estimates.

We have restructured, extended, and clarified the discussion in Sects. 3.1 and 3.2.

For the discussion, I suggest, again, focusing on the goal of the study. If the goal is CH4 production and transport, then some simple calculations about the impact of the parameters measured here on the transport and emissions would be helpful. As it is, the discussion reads like a collection of separate paragraphs where the importance of more complex network parameters such as tortuosity or top-bottom betweenness centrality for CH4 production and gas transport are left unexplained, while some relatively obvious results, like that higher porosity lead to higher connectivity (lines 436-444) are discussed in length.

The aim of the study is to qualitatively characterize the structure of peat and the changes in the structure with depth using methods adopted from complex network theory. Issues related to dynamic properties and processes and to comparison with experimental methods related to gas diffusion efficiency in peat will be the subject of further studies. We analyze how different network metrics are capable of describing network connectivity and how (1) porosity and connectivity measures and (2) local connectivity measures and global connectivity measures are related in a complex peat pore network. We state that the connection between porosity and average local connectivity may not always be straightforward in peat pore networks. Local connectivity may be high even if porosity or global connectivity is low. This might be the case, e.g., if the connected pore space does not extend over the whole network domain. We found that the applicability of local connectivity metrics to describe global network properties is dependent on the shape and topology of the connected pore space.

Geometrical tortuosity is, by definition, a direct proxy for network transport efficiency as it illustrates the tortuosity and length of flow paths through a network. Centrality measures (closeness centrality, betweenness centrality) are more advanced metrics adapted from complex network theory. Using geometrical tortuosity as a proxy, we could estimate how well the centrality measures and other network theory metrics are able to describe the transport properties of a network. Top-bottom betweenness centrality is, as it is defined in our study, an adaptation of the network measure betweenness centrality. Because of its direction-based definition, top-bottom betweenness centrality is more directly related to transport processes through a network than the other network theory metrics in the study. We have discussed further on this role of geometrical tortuosity in the performance analysis and on the nature and relevance of top-bottom betweenness centrality in the Discussion (Sect. 4.2 in the revised manuscript).

**Specific comments:**

Line 14: What was the hysteresis in relation to? What were the parameters that showed the hysteresis behavior?

The hysteretic behavior was seen as the difference in the volumetric water content of a sample at a specific soil water potential between drying and wetting processes. The effect is also reflected in the volume fractions of air-filled pore networks in Fig. 9. Under the same pressure conditions, the volume fraction of water in the pore space was considerably smaller during wetting than during drying. We have clarified the context in the Abstract (L19).

Line 23: Please be more specific about how peatlands modulate or are modulators of hydrological and biogeochemical cycles. Why are they globally important?

Vast quantities of water are stored in peatlands, and they may either accelerate or attenuate flooding in watersheds (Holden 2005). Peatlands store one third of global soil carbon (Kang et al. 2018), and boreal and subarctic peatlands store 300-400 Pg carbon as peat (Turunen et al. 2002), which corresponds to nearly half of the carbon currently held in the atmosphere (Limpens 2008). A large proportion of dissolved organic carbon in freshwater ecosystems originates from watersheds with peatlands (Kang et al. 2018, Asmala et al. 2019). We have clarified and extended the reasoning at the beginning of the Introduction (L25–29).

Line 28: The line starts with repetition.

We have reformulated the related sentences and removed the repetition (L31).

Line 39: Capability instead of characteristic?

In soil science, the functional relationship between soil water potential and soil moisture is called, e.g., the soil-moisture retention curve, the soil-moisture release curve or the soil moisture characteristic (Hillel 1998); the water release characteristic (Townend et al. 2001); the soil-water retention curve (van Genuchten 1980); or the water retention characteristic (Raats 2001).

Line 96: Why was the diameter of the sample only measured to determine shrinkage? Where the samples all the time in the plastic tubes? Why was the vertical shrinkage not considered?

As the manuscript text states (L106), also the height of the sample was measured and the vertical shrinkage was considered. In most samples, vertical shrinkage was greater than horizontal shrinkage. The samples were not removed in the acrylic tubes during the water retention measurement process.

Line 104: How good is the value of 1500 kg m-3 for particle density for these samples? Does it accurately represent all the sample depths?

The value of 1500 kg m-3 is a widely used reference value for particle density in boreal peat soil (Paavilainen and Päivänen 1995). The variation in particle density with depth (classified by bulk density) is very small in peat (Päivänen 1973).

Line 106: The assumption that air-filled porosity was zero in the saturated state is probably not quite true. Please give an estimate for how much that might be off in your samples and what kind of an error it would lead to in your results.

The mass of the saturated sample was not used in the determination of the total porosity, as the total porosity was calculated using the measured dry mass and estimated particle density (Eq. 2). Therefore, the assumption that air-filled porosity was zero in the saturated state did not affect the air-filled porosity calculations at other matric potentials. The assumption was only reflected in the fitting of the van Genuchten model, where we separately state the saturated water content was assumed to equal the total porosity that we had estimated. This assumption was also used in Weiss et al. (1998). In addition, the assumption was used in Fig. 4 as the air-filled porosity value at -0.1 kPa matric potential, but this value is not essential in the analysis of the water retention properties presented in the figure. Because the assumption is used only for illustrative purposes in Fig. 4, we have removed the statement from this section and stated the choice and given reasons for it in the caption of Fig. 4.

The deviation from total saturation is naturally dependent on the method of sample saturation. An estimated range of the air-filled porosity in the saturated state in laboratory conditions is about 2...3% (Päivänen 1973). The larger the deviation from zero air-filled porosity in the saturated state, the smaller are the values of the parameters  $\alpha$  and n in the optimal fit. However, because the parameter values are not of primary importance in the study, we consider that a more thorough error analysis would be beyond the scope of the study.

Eq 3. Why is this equation needed? Why is not the calculation using eq 1 not enough for the comparisons? How was this equation fitted to data? What were the fitted variables that were measured, and how were they measured?

The water retention properties of soil are characterized by the water retention curve, that is, the functional dependence of volumetric water content on soil matric potential. The functional form of the dependence is used in process-based models to characterize the properties of soil in different wetness conditions. The van Genuchten parameterization of the water retention curve (van Genuchten 1980) is widely used in soil science, and the van Genuchten parameters can be used to classify the soil properties and illustrate the characteristics of a specific soil sample.

The fitted variables were soil matric potential  $\Psi$  and volumetric water content  $\vartheta$ . The determination of  $\vartheta$  is described in Sect. 2.2 and the formula is presented in Eq. 1. The soil matric potentials were equivalent to the pressures applied to the peat samples pressure plate apparatus. The fitting was performed for the average values of  $\vartheta$  of the samples from each depth by applying the Levenberg– Marquardt algorithm using the Python library SciPy designed for scientific computing. As the result of the fitting, a function  $\vartheta(\Psi)$  was obtained by determining values for the parameters  $\alpha$  and n in Eq. 3. Line 113-114: Why is it important to tell that the pressure range was so narrow that the fitting procedure needed to be modified? Why was a larger pressure range or a more appropriate fitting algorithm not used?

The study was focused on macroporosity (pore diameter > 100  $\mu$ m) because smaller pores are usually water-filled in peatlands where the water table depth is generally less than 1 m; thus, gas diffusion in air occurs only in macropores. In addition, it is the pore size range that can be investigated through  $\mu$ CT imaging so that the sample size remains reasonably large. Macropores are supposed to be emptied of water at a matric potential of -10 kPa; therefore, the retention measurements were performed up to -10 kPa potential. The shape of the van Genuchten retention curve is such that this potential range does not cover the whole shape, therefore the end points must be fixed. This also fixes two of the model parameters, the saturated water content and the residual water content. The main focus of the work was not to study the retention properties of peat on a larger range, and the retention model fit was a supplementary result in the study. We have elaborated the description in the text (L120–121).

Line 118: Please move "noninvasively" to the end of the sentence.

We have corrected the word order (L126).

Line 123-125: Please give an error estimate for what this darkening might do to your results.

We state in the Discussion (Sect. 4.1) that the darkening of the  $\mu$ CT images due to image reconstruction defects near the top and bottom of the sample increased the void fraction slightly in all binary images. It is very difficult to give a quantitative estimate of the effect of the darkening on the obtained void fractions of the images. This error only concerns the air-filled porosity estimates of the whole samples presented in Fig. 1 because the top and bottom regions were excluded from the pore network domains.

Line 136: How did you deal with that the contrast between air-filled and water containing regions or organic materials was low? How did this affect your results? What kind of and how large an error could it cause?

We used a standard binary segmentation algorithm, Otsu's method, to perform the solid-void classification. This way the segmentation result contains a minimal amount of operator bias. The fraction of void space in the image may have been over- or underestimated depending on if the threshold intensity was high or low, respectively. The magnitude of the error is very difficult to estimate. Further, each stage in the workflow from  $\mu$ CT imaging to pore network generation is prone to contain some inherent discrepancy. We find that a thorough analysis of image segmentation and the sources of discrepancy in it is beyond the scope of this study. We, however, have returned to this issue qualitatively in Discussion.

L361–365: "In addition, the low intensity contrast between air and water or organic matter in the  $\mu$ CT images obscured the determination of the boundaries of the air-filled regions. Generally, this may have had either an increasing or a decreasing effect on the determined air-filled volume. For example, the global solid-void classification method seems to have overestimated the void volume in some of the images of the top layer samples."

Line 154: Why did you use the sum of distances? It is hard to visualize how this calculation what performed and what it means.

The centroids of adjacent pores and the centroid of the throat between them are usually not collinear, and the linear distance between the pore centroids is therefore smaller than the length of the route from the centroid of pore 1 through the connecting throat to the centroid of pore 2 (see Fig. AR1). We have clarified the definition.

L164–166: "Because the centroids of adjacent pores and the centroid of the throat between them were usually not collinear, the distance d between adjacent pores was determined as the sum of the distances between the centroids of each pore ( $p_1$  and  $p_2$ , respectively) and the centroid of the throat (t):  $d(p_1, p_2) = d(p_1, t) + d(t, p_2)$ ."

Centroid of pore 1

Figure AR1. Schematic 2D illustration of two pores and a throat. The linear distance (lighter gray) is smaller than the length of the actual route between the pore centroids through the throat (darker gray).

Line 158-160: Does this mean that the shrinkage of the samples was not taken into account here at all?

It was assumed that the peat sample filled the acrylic cylinder completely in the initial, saturated state. The void volume, that is, the volume inside the boundaries of the cylinder not occupied by the peat matrix, was completely filled with water (and a little amount of air). When the sample started to dry and compress, the pore spaces inside the sample started to collapse and become smaller. This means that the air-filled regions inside the sample were effectively transferred into the space between the sample and the cylinder boundaries. The images were taken at -10 kPa matric potential when the sample was shrunk. If it is assumed that the solid matter volume remained constant during the shrinkage process, the volume of the non-solid matter inside the boundaries of the cylinder also remained constant. If the volume fraction of air-filled space is now determined with respect to the total volume of the cylinder, the shrinkage of the samples will be taken into account. We have clarified in the text (L172–174) that this procedure ensures that the effect of shrinkage is included.

Line 164: What does the "largest of these clusters" mean? How was size defined? How does taking this part under analysis solely affect the interpretation of the porosity and connectivity results from the samples? Can this size be somehow standardized between samples to make them comparable? Can there be more than 1 large enough network in a sample that all of them should be considered?

The size of the pore clusters was defined by their volume. The largest connected cluster was the cluster that extended through the network domain in the vertical direction; other clusters were not connected to both open ends of the network domain. As is seen from the results in Sect. 3.1, the largest cluster contained on average about 90% of the total pore space volume. Therefore, the transport processes that occur in the pore system are restricted only to the largest cluster, that is, the connected network. The connected network is the relevant space when assessing the transport properties of the pore system. The impact of this limitation is discussed in Results in Sect. 3.1: the network volumes were smaller than the corresponding image void space volumes. If there were two or more connected networks that extended through the whole domain, all of them should be included in the analysis, but this was not the case in this study. We have emphasized in the text (Sect. 3.1, L267–268) that the largest pore clusters were the only pore clusters that extended through the network domain in each pore system. We have also included the assumption that the largest cluster is the only one that extends through the domain in the definition of pore network in Sect. 2.7.1 (L176).

Line 202: See previous comment. How does the defined region affect the connectivity analysis. How can you account for non-connectivity in a standardized sample size?

Due to computational limitations, we needed to assume that the selected subregion was representative of the whole sample and that the generated network was representative of the network that would have been generated using the whole sample as the domain. The largest cluster contained on average about 80% of the total pore space volume of the cubical-domain networks and was always the only cluster that extended through the network domain in the vertical direction. Therefore, the smaller pore clusters would not have taken part in transport processes through the network. We have clarified the reasoning for the selection of only the largest cluster for network analysis and the dominance of the volume fraction of the largest cluster in the text (L216–218).

Line 234: Please explain more clearly how the one-way anova analysis was performed. Were all the depths analyzed simultaneously or did you perform pairwise tests between each. Would a multi-way anova with interactions included or a version of linear models give you more comprehensive results?

We determined if (1) water retention characteristics differed between depths, (2) if pore sizes differed between depths, and (3) if network metrics differed between depths. We thus analyzed the influence of one categorical variable (depth) of one continuous dependent variable at a time. There was only one categorical variable, and the use of a multi-way ANOVA was therefore not possible. The main interest was the variability of each of the listed properties with depth. There may have been interactions between variables, but the number of degrees of freedom was too low for a more thorough analysis. Also, obvious correlations between independent variables (especially water

retention characteristics and pore sizes) may weaken the power of linear models. We have clarified the description of ANOVA in the text (L249–250).

Line 242: Figure 3 actually shows a consistent underestimate of porosity by the model.

The void fractions obtained from the images were indeed slightly lower than their measurementbased counterparts. We have changed the wording (L258–259).

Line 254: Does water content here refer to the original samples, or saturated samples or dried samples, and at what depth?

As it is stated in the text (L273–274), water content in all studied matric potentials was clearly lowest in the top layer and highest in the deepest studied peat layer. Water content refers to the samples in each state during the retention measurement. Water content was lowest in the top layer in the -1 kPa conditions, and it remained lowest in all the progressively drier (-3 kPa, -6 kPa and -10 kPa) conditions. Correspondingly, water content remained highest in the deepest layer in all matric potential conditions. The water content of the original samples was not measured, and the water content of a dried sample is zero by definition.

Line 260: The hysteresis effect seems to be only a modeling result. Is that a feature of the model, or is there real experimental evidence for that this hysteresis happens?

The phenomenon that different water saturations are reached at the same capillary pressure, depending on whether the wetting fluid (wetting/imbibition/sorption) or the nonwetting fluid (drying/drainage/desorption) is the displacing phase, is a well-known issue both in porous media in general (see Dullien 2000) and in soil (e.g., Haines 1930, Poulovassilis 1962). The experimental determination of the sorption curve is much more difficult than the determination of the desorption curve, and therefore it is seldom done (Hillel 1988). For most practical applications, only the desorption curve is used and the effect of hysteresis is ignored (Townend et al. 2001). Because air-filled porosity is obtained by subtracting the volumetric water content from total porosity, hysteresis of volumetric water content translates directly into hysteresis of air-filled porosity.

Figure 6: I think combining the cumulative volume figures to one figure where each depth is described by one average line and error bars (or error shading) would be a mode effective way to present this data. It is not necessary to present data for individual samples here.

We believe that the figure is most illustrative in its current form. The panels c-e in Fig. 6 show the large variation of both the total network volumes and the sizes of the largest pores between samples.

Lines 371-380: How do the results of this study relate to this text?

We showed that the volume of air-conducting pathways in peat is dependent on its wetness and that the volume of the connected pore space decreases with increasing water content. An increasing fraction of the peat pore space therefore becomes isolated from the atmosphere, and these locations become favorable for anaerobic processes. We have restructured the Discussion and stated more explicitly that we are assessing conceptual implications of the pore network structure of peat for methane-related processes in Sect. 4.3 in the revised manuscript.

Line 442: Can anything else happen in networks than shorter pathways leading to more alternate routes?

When local network connectivity was high (implying that a random pore was connected to several other pores), the average traveling distance from a random pore to another random pore was relatively short. When pore connectivity is higher, there are more alternative routes available between two random pores, some of which may be spatially rather straight. Therefore, a higher number of alternate routes is the cause and a shorter pathway is the effect in this case.

Lines 454-459: How is this related to gas transport of CH4 production?

The ratio of the average clustering coefficient and the average coordination number may give insight on the efficiency of gas transport through a peat pore network. As it is stated in the text, the ratio of the two metrics may reflect the abundance of long and narrow pore conduits in the network, which may govern the number of alternative transport routes through the network. A low number of alternative routes (a low ratio) may lead to a rapid decrease of gas transport if one of the routes gets blocked, and a high ratio may enhance the resilience of the network to local blockages. We have clarified the text.

L428–434: "The ratio between the two local connectivity measures may reflect the structure of a pore network in regard to gas transfer efficiency. A high ratio of the network average clustering coefficient to the network average coordination number indicates that the number of three-pore loops is relatively high and may also imply that even larger clusters of interconnected nearby pores are abundant in the network. This may be reflected as a high gas transfer capacity and good resilience to disturbances. By contrast, a low ratio may result from a high abundance of long nonbranching pore conduits in the network, which may diminish the amount of alternative transport routes within the network and lead to a rapid suppression of gas transport if pores become clogged."

Line 494-495: How do you get to the conclusion that orientation in diffusion paths does not change transfer towards atmosphere?

We meant that the diffusion paths are oriented in such a way that gas transfer towards the atmosphere is neither promoted nor restrained in deeper peat layers. We have corrected the sentence.

L472–473: "According to our results, the orientation of diffusion paths in pore networks is such that it does not promote or restrain gas transfer towards the atmosphere in deeper, more degraded peat layers."

**References**

Asmala, E., Carstensen, J., and Räike, A.: Multiple anthropogenic drivers behind upward trends in organic carbon concentrations in boreal rivers, Environ. Res. Lett., 14, 124 018, 2019. Dullien, F. A. L.: Capillary and viscous effects in porous media, in: Handbook of Porous Media, edited by Vafai, K., pp. 53–111, Marcel Dekker, New York, NY, 2001. Haines, W. B.: Studies in the physical properties of soils. V. The hysteresis effect in capillary properties, and the modes of moisture distribution associated therewith, J. Agr. Sci., 20, 97–176, 1930. Hillel, D.: Environmental Soil Physics, Academic Press, San Diego, California, 1998.

Holden, J.: Peatland hydrology and carbon release: why small-scale process matters, Philos. Trans. R. Soc. A, 363, 2891–2913, 2005.

Kang, H., Kwon, M. J., Kim, S., Lee, S., Jones, T. G., Johncock, A. C., Haraguchi, A., and Freeman, C.: Biologically driven DOC release from peatlands during recovery from acidification, Nat. Commun., 9, 3807, 2018.

Limpens, J., Berendse, F., Blodau, C., Canadell, J. G., Freeman, C., Holden, J., Roulet, N., Rydin, H., and Schaepman-Strub, G.: Peatlands and the carbon cycle: from local processes to global implications – a synthesis, Biogeosciences, 5, 1475–1491, 2008.

Minkkinen, K. and Laine, J.: Effect of forest drainage on the peat bulk density of pine mires in Finland, Can. J. For. Res., 28, 178–186, 1998.

Paavilainen, E. and Päivänen, J.: Peatland forestry: Ecology and principles, Ecological Studies 111, Springer-Verlag, Berlin, Germany, 1995.

Päivänen, J.: Hydraulic conductivity and water retention in peat soils, Acta For. Fenn., 129, 1–70, 1973.

Poulovassilis, A.: Hysteresis of pore water, an application of the concept of independent domains, Soil Sci., 93, 405–412, 1962.

Raats, P. A.: Developments in soil–water physics since the mid 1960s. Geoderma, 100, 355–387, 2001.

Rezanezhad, F., Price, J. S., Quinton, W. L., Lennartz, B., Milojevic, T., and Van Cappellen, P.: Structure of peat soils and implications for water storage, flow and solute transport: A review update for geochemists, Chem. Geol., 429, 75–84, 2016.

Townend, J., Reeve, M. J., and Carter, A.: Water release characteristic, in: Soil and Environmental Analysis: Physical Methods, edited by Smith, K. A. and Mullins, C. E., pp. 95–140, Marcel Dekker, New York, NY, 2001.

Turunen, J., Tomppo, E., Tolonen, K., and Reinikainen, A.: Estimating carbon accumulation rates of undrained mires in Finland – application to boreal and subarctic regions, Holocene, 12, 69–80, 2002. van Genuchten, M. T.: A closed-form equation for predicting the hydraulic conductivity of unsaturated soils, Soil Sci. Soc. Am. J., 44, 892–898, 1980.

Weiss, R., Alm, J., Laiho, R., and Laine, J.: Modeling moisture retention in peat soils, Soil Sci. Soc. Am. J., 62, 305–313, 1998.

**Referee #2**

The manuscript is a carefully detailed and well described study on pore network analyses of peat soils with a depth differentiated view. The aim is to use results of pore network analyses to help explain methane production potential in these environments. The level of language is very high, the text flow is very good, and it was a great pleasure to review. Although I am not that familiar with micro CT, image analyses and pore network simulations, the results are feasible, while not super surprising, as would be expected from the literature. Still, the manuscript is novel in its approach and comparisons.

I was left a little bewildered, what had happened to the discussion of methane (L371-383 gives some general statements on the importance and general conceptual discussion), in particular, since it was not explicitly modelled nor were the results contrasted against methane flux measurements from the field.

My recommendation is that, now with some distance, the authors re-assess the paper and whether or not the self set aims are fully fullfilled. Perhaps it might be a good idea to focus a little more on the strong technical part of the manuscript and limit the study a little more to the descriptive nature, and then discuss the methane production and diffusion from this vantage points in a more speculative manner. After all, the emerging methane emissions are mostly explaned by a conceptual model.

The aim of our work is to study the capability of complex network theory methods and metrics to characterize the physical structure of peat and its pore space and to qualitatively assess the obtained metrics and correlations between them in relation to gas transfer processes in peat. The study also shows the physical existence and evolution of isolated air-filled pockets and illustrates their capability to control the production and emission of gases, and methane in particular, from peat. A more thorough comparison between the pore network modelling method and experimental methods related to gas diffusion efficiency in peat will be the subject of further studies. We have revised and reformulated the study aims in the last paragraph of the Introduction (L75–79). We have also restructured the Abstract, Discussion, and Conclusions so that the conceptual nature of the assessment of methane-related processes is now indicated more clearly.

**Major concern**

What does this study have to do with methane emissions? This remained unclear to me. Either, the scope of the study should be a more quantitative

The study is part of a larger project that aims to develop tools for studying and modeling methane emissions from peat. The study aims to create a conceptual basis for the description of processes related to methane generation and its atmospheric emissions. The methods presented and used in this study to assess the structure and connectivity of peat pore space can later be applied to describe the processes related to gas transfer and biogeochemical processes in peat. Specifically, we aim to qualitatively evaluate the potential to use the pore network characteristics revealed by µCT imaging

and the complex network theory metrics calculated for these networks for assessing gas transfer and processes related to organic matter degradation in peat. Methane generation generally requires anoxic conditions, and studying the development and evolution of regions prone to anoxia in peat illustrates the capabilities of peat for methane generation. We have revised the aims of the study and restructured the Abstract, Discussion, and Conclusions so that the conceptual nature of the assessment of methane-related processes is indicated more clearly.

**Minor concerns**

L5 How can the formation of anaerobic pockets be conceptualized in a pore network approach? This is left unaddressed.

The formation and evolution of pore regions isolated from the atmosphere during peat wetting can be assessed and illustrated using pore network modeling. Performing a porosimetry simulation in a pore network gives a description of the gradual change of the number of air-filled pores and the development of isolated pore clusters. Therefore, a pore network modeling approach offers a way to conceptualize the physical basis of the formation of anaerobic pockets. This is illustrated in the modeling results in Sect. 3.5. We have clarified the issue and reformulated the sentence in the Abstract.

L5–6: "The evolution of the pore space that is connected to the atmosphere can also be conceptualized through a pore network modeling approach. Pore regions isolated from the atmosphere may further develop into anaerobic pockets, which are local hotspots of  $CH_4$  production in unsaturated peat."

L42-43 Here peat specific citations should be made (e.g. Hayward and Clymo (1982), Weber et al., 2017), after all, the suggest a pore size distribution.

We have added the suggested references (L49).

L53 like previous comments.

We have added the suggested reference (L62).

L 90 ff section 2.2: was the VGM model fitted to the averages of the replicates from each depth, respectively? Just a minor information to add.

Yes. We have clarified the description (L119).

L94 Sentence not needed, but not harmful, either.

We have kept the sentence for clarity.

L97 in -> at

We have corrected the preposition (L107).

L105 This is a rather bold statement. If samples were not saturated under co2 environment of vacuum, my impression is this is not actually correct. Please do not use this assumption.

We did not use the assumption in the calculation of the air-filled porosity of the samples. The mass of the saturated sample was not used in the determination of the total porosity, as the total porosity was calculated using the measured dry mass and an estimated, constant particle density (Eq. 2). Therefore, the assumption that air-filled porosity was zero in the saturated state did not affect the air-filled porosity calculations at other matric potentials. The assumption was only reflected in the fitting of the van Genuchten model, where we state that the value of the saturated water content parameter was assumed to equal the total porosity that we had estimated. In addition, the assumption was used in Fig. 4 as the measured air-filled porosity value at -0.1 kPa matric potential, but this value is not essential in the analysis of the water retention properties presented in the figure. The assumption of full saturation is also consistent with the fact that the pore network was, by definition, filled with water at the initial state of the porosimetry simulation.

Because the assumption is used only for illustrative purposes in Fig. 4, we have removed the statement from this section and stated the choice and given reasons for it in the caption of Fig. 4.

L118 quantify the sample: What is meant by this? Please specify.

We meant that the structural properties, such as pore space volume and pore size distributions, can be assessed quantitatively. We have clarified the sentence (L126–127).

L122 is resolution meant, here?

In tomography, voxel size and (spatial) resolution are different concepts. Voxel size is the spatial dimension represented by one image element. Spatial resolution is related to the size of the smallest distinguishable feature in the image, and it is roughly twice the voxel size in microtomography.

L122 if this is the resolution, then the real micropores

L 290: This pattern can, again, be observed in Weber et al. (2017), but also in other sources in the literature. I suggest adding citations, here.

Disappearance of the largest pores and a smaller spatial variability of the pore size distribution in greater depths observed by Weber et al. (2017) can be inferred from Fig. 8a that shows that the variation between the total pore network volume and the number of pores was largest in the top layer and decreased with depth. However, Fig. 8a only indicates that the spatial variation of average pore volume is largest in the top layer. We prefer not to use citations in the Results section. We return to the changes in pore space properties with depth in the Discussion (Sect. 4.3 in the revised manuscript). We have further elaborated this issue and referred to the literature in the Discussion.

L520–523: "In addition, the variation of porosity and the variation of average pore volume between the samples were largest in the top layer (Fig. 8a). Increasing degree of decomposition deeper in the peat profile also results in the homogenization of peat and a smaller spatial variability of the pore size distribution (Weber et al., 2017)."

L384-385 repetition of introduction

We would prefer to keep this sentence (L491–492) for continuity and as a transition to the subject of the paragraph.

L393: a bit of a leap of faith or perhaps a rather general statement: the simulation provide support for the conceptual understanding to be ok. I think these hard earned results should perhaps be discussed a little more in acknowledgement to this (although I detect absolutely no oversell, here).

We showed that the evolution of isolated air-filled pore space with respect to changes in matric potential was considerably different in drying and wetting. The air-filled pores with no connection to the atmosphere are prone to anoxia. This may give reason to suppose that also the development of anaerobic pockets differs in wetting and drying. The susceptibility of unsaturated peat to anoxia and methane generation is discussed in more length in the previous paragraphs of the section. We have elucidated and justified the argument further in the text.

L500–504: "Our drainage and imbibition simulations indicated that hysteresis may affect the evolution of the fraction of peat pore space that is isolated from the atmosphere. The volume fraction of the air-filled pore space connected to the atmosphere at a specific matric potential was considerably smaller during drying than during wetting. This may further provide support for the conception that hysteresis may also affect the formation and destruction of anaerobic pockets and the temporal CH4 dynamics in the unsaturated layer."

L425: I see the potential to discuss the obtained pore space characterizing numbers in the light of other peoples results re: methane transport, or any other gas transport in particular. If this does not exist, I suggest to scope this section a little more carefully.

We use geometrical tortuosity as a proxy for diffusion efficiency and assess the applicability of network theory measures to characterize the transport properties of a network by comparing them with geometrical tortuosity. Geometrical tortuosity is, by definition, a direct proxy for network transport efficiency as it illustrates the tortuosity and length of flow paths through a network.

Centrality measures (closeness centrality, betweenness centrality) are more advanced metrics adapted from complex network theory. Using geometrical tortuosity as a proxy, we could estimate qualitatively how well the centrality measures and other network theory metrics are able to describe the transport properties of a network. To our knowledge, there are no previous experimental studies on the use of complex network theory metrics for gas transfer in natural porous media. We have reformulated the title and the scope of the section, which is now Sect. 4.2, and emphasized the role of geometrical tortuosity as a proxy in performance analysis.

L395-401: "Van der Linden (2016) highlights the need for multiscale measures of pore space topology and connectivity and the relationship of these descriptors to macroscopic transport processes in a porous medium, such as peat. Geometrical tortuosity is a structural characteristic of a porous medium (Clennell, 1997). It gives an estimate of the average path length through a network in a specified direction, which is a direct proxy for network transport efficiency. Therefore, it can be used as a benchmark measure of the applicability of different complex network theory metrics to characterize the transport properties of a network and to estimate the efficiency of macroscopic transfer processes in a network. We compared several kinds of network metrics with the gas transfer capacity estimated by geometrical tortuosity. [...]"

L448 mark you, many of the dead end pores might not have been resolved due to the micro CT technique. Perhaps this should be contextualize, too.

The paragraph concerns the structure of a pore network and the characteristics and possible weaknesses of centrality measures in general, with no reference to a specific realization of a network or the relation of a network to a real porous object it represents. We think that a reference to the specific network is not relevant in the context of this paragraph (L420–427).

L486: How can a relative quantitative measure (anisotropy) determine the efficiency (a qualitative) absolute measure of gas diffusion.

We did not mean the quantitative performance measure named efficiency but the capability of gas diffusion in different directions. We have changed the wording.

L463–464: "The structural anisotropy of peat, characterized by the difference between vertical and horizontal geometrical tortuosity, can be used to estimate the diffusion capability of a gas in different directions through peat."

**References**

Hayward, P. M. and Clymo, R. S.: Profiles of Water Content and Pore Size in Sphagnum and Peat, and their Relation to Peat Bog Ecology, Proceedings of the Royal Society B: Biological Sciences, 215, 299–325, https://doi.org/10.1098/rspb.1982.0044, 1982.

Weber, T. K. D., Iden, S. C., and Durner, W.: A pore-size classification for peat bogs derived from unsaturated hydraulic properties, Hydrol. Earth Syst. Sci., 21, 6185–6200, https://doi.org/10.5194/hess-21-6185-2017, 2017.

---

## Author Response (AR2)

**bg-2021-259 Technical corrections**

We would like to thank the editor and the referees for the effort and time they put into reviewing our manuscript. We have considered and addressed the comments in the revised manuscript and corrected or otherwise clarified the issues that the editor and the referees had raised. In the following list, editor the referee comments are written in bold characters, whereas our responses are shown in regular characters. The line numbers in the referee comments in our responses refer to the original version of the manuscript, whereas the line numbers in our responses refer to the revised manuscript.

**Response to Associate Editor Decision**

Thanks for your revision, which was re-read by both reviewers, who both generally agree that their main concerns have been addressed. Reviewer 1 remains concerns by the lack of focus, and length, of the results and discussion, and provides many helpful suggestions on improving these parts (I would add, re the discussion: try not to restate results!). I agree that these sections are quite long and could be usefully streamlined and on occasion restructured; doing so will help readers and improve the impact of your work. For this reason, please consider and address all of R1's remaining comments, and look for opportunities to tighten the text within reason.

We consider that an extensive discussion on network metrics from the viewpoint of peat structure in Sect. 4.2 is relevant to the aims of the study. The third aim of the study is to assess the capability of complex network theory metrics to describe the physical structure of peat pore space. Also, the title of Sect 4.2 is "Network connectivity metrics related to peat structure and gas transport". Tightening the text would cut down the discussion on one of the main aims of the study. In the revision of the manuscript, we have shifted the main focus from assessment of methane transport capability more towards the description of the physical peat structure by the network metrics as it was suggested by the referees in the first revision round. Because we have introduced the metrics such as closeness centrality and betweenness centrality to the analysis of peat pore structure, we consider that discussion on the applicability of these metrics is justified.

We think that referring to results in question in the Discussion functions as an introductory sentence for the discussion in each paragraph and thus makes the text more readable and comprehensible. This is justified because the network metrics is a new concept in this field of science.

**Response to comments from Referee #1**

This manuscript has significantly improved in both clarity and motivation after the revisions. The methods are now mostly described with enough detail, and the focus of the manuscript is clearer. This said, however, some reorganization and rewording in the text could significantly increase the impact of the study. The weakest part is the results and discussion where the focus of the study to analyze peat structure to understand gas transport is sometimes completely lost. To make the logic behind the study and the conclusions understandable for a reader who may not be an expert in this field, explaining the meaning of the results in the context of gas transport early on would be helpful. Here below some specific suggestions and comments to try to help with this work.

The focus of the study is not only to analyze peat structure to understand gas transport, but also to analyze the applicability of complex network theory metrics to the description of the physical structure of peat, and this is also discussed in Sect. 4.2 in the Discussion, named as "Network connectivity metrics related to peat structure and gas transport". We discuss the reasons behind our finding that local and global connectivity metrics do not always properly describe the structure and properties of a peat pore network, which is stated in L412–413 in the manuscript and mentioned also in the Abstract. We think that this is an important issue in the study.

**Specific comments:**

Line 1: Here it would be helpful if processes leading to CH4 production and transport were separated, so that the reader would get insight why oxygen transport suddenly controls CH4 emissions.

We have clarified the sentence.

L1–2: "The production and emission of  $CH_4$  are strongly influenced by the diffusion of oxygen into the soil and of  $CH_4$  from the soil to the atmosphere, respectively."

Line 14: Here explaining whether the channel structure affects CH4 transport vs. production would be helpful to guide the reader in how to think about the results.

Here, the discussion on channel structure is focused on the effect on gas transport in general, not specifically on the effect on CH4. We consider that we already state explicitly in the sentence that channel structure affects gas transport.

Line 18: Please delete word "global". The reader does not yet know what that refers to in this context.

**Corrected.**

Line 45-50: Please make the connection between gas transport and water retention clear here. Why is water retention an important feature when trying to understand gas transport? This could be done by moving text from lines 60-65 to be earlier.

We consider that the connection between the water retention characteristic and gas transport dynamics is discussed thoroughly in the present paragraph. To make the text clearer, we have moved the first part of the sentence from L62 to the previous paragraph and rephrased the remaining text in L62–63. Further sentence restructuring would break the current structure and flow of the Introduction. We have also clarified the sentence starting from L44.

L44–45: "Water retention characteristic is a fundamental soil property that links soil structure to water dynamics, gas transport, redox conditions, and many accompanying biogeochemical processes (Bachmann and van der Ploeg, 2002; Lepilin et al., 2019).

Line 98: "placed" instead of "located"

Corrected.

Lines 116-120: Please explain why eq (3) was used and why eq (2) was not enough? It seems strange that eq (3) was used even if the fit was not good. It is also unclear which of these equations was used to produce the data in the figures.

The water retention properties of soil are characterized by the water retention characteristic, that is, the functional dependence of volumetric water content on soil matric potential. Water retention characteristic is a fundamental soil property that sets the foundation for water content and energy state and therefore water transport in soil, and essentially affects gas transfer. The functional form of the water retention characteristic is required in all process-based models. Several equations with empirically determined parameters have been proposed to describe the functional form of the water retention characteristic. The van Genuchten parameterization of the water retention curve is widely used in soil science, and the van Genuchten parameters,  $\alpha$  and n, can be used to classify soil properties and illustrate the characteristics of a specific soil sample. We have added a short introductory sentence to the description of the model.

Equation (3) was not used for data presentation. The fitted van Genuchten model parameters are only presented in Table 1 for reference and for comparison purposes. We have added references to the equations used in the calculations to the captions of Figs. 3, 4, and 5.

L116–118: "Many empirical models have been constructed to describe the functional form of the water retention characteristic (Hillel, 1998). For comparison purposes, water retention properties were also characterized using the widely used van Genuchten model (van Genuchten, 1980)".

Line 124: Delete "In short"

Corrected.

Line 125: "With this method..."

Corrected.

Line 126: "obtain" instead of "get"; replace "and" by a comma after "noninvasively"

Corrected.

Line 134: How did you do the reconstruction? How was background subtracted from the CT data?

We have added information on the image reconstruction.

L133: "The conversion of the radiographs to grayscale images was performed with the GE image reconstruction software datos |x."

Line 141: How did you deal with the separation of the acrylic, please explain briefly so that the reader gets an idea if this method is reasonable.

The separation was performed in a very simple way: a cylindrical region (diameter 1000 voxels) was cropped, that is, selected from the cubical image with horizontal dimensions of 1000 x 1000 voxels. The inner diameter of the acrylic cylinder was 50 mm (1000 voxels), so the cylinder was cut out of the image. We have clarified the wording.

L143–144: "A cylindrical peat volume (height 1000 voxels, diameter 1000 voxels) excluding the acrylic cylinder was selected using PoreSpy."

Line 146: How did you deal with the challenge that the contrast was low? What error would that induce in your results? Please give an estimate.

The fraction of void space in the image may have been over- or underestimated depending on if the threshold intensity determined for the low-contrast images was too high or too low, respectively. The magnitude of the error is very difficult to estimate. Further, each stage in the workflow from  $\mu$ CT imaging to pore network generation is prone to contain some inherent discrepancy. We find that a thorough analysis of image segmentation and the sources of discrepancy in it is still beyond the scope of this study. We address this issue qualitatively in Discussion: "In addition, the low intensity contrast between air and water or organic matter in the  $\mu$ CT images obscured the determination of the boundaries of the air-filled regions. Generally, this may have had either an increasing or a decreasing effect on the determined air-filled volume. For example, the global solid–void classification method seems to have overestimated the void volume in some of the images of the top layer samples."

Line 70: Why was it assumed that the sample surfaces were level with the ends of the cylinder? Did you not see that from your samples?

The sample surfaces were probably not totally smooth at the length scale of 50  $\mu$ m but there was evidently some vertical fluctuation. We have clarified the sentence.

L172–174: "In the calculations, it was assumed that the sample surfaces had been totally smooth and in level with the ends of the acrylic cylinder in the initial, saturated state."

Line 176: "defined" instead of "assumed"?

In the context of this study and the terminology determined in this section, we set the assumption that there is only one pore cluster extending through the sample domain and refer to it as the pore network. Generally, the number of this kind of clusters is not restricted to one. If there are several this kind of clusters in the sample domain, all of them should be included in the analysis. However, this was not the case in this study.

Line 179: "The volume of the total pore size was..."

Corrected.

Line 184: Does the OpenPNM package use eq 2 or 3 in the calculations? It is hard to understand here what was done with which model of equation and which figure shows results by which method. It is also unclear why all these methods were needed.

The drainage percolation simulation does not use the equations presented in Sect. 2.2, as it simulates the gradual filling of the pore network governed by Eq. (4). The percolation algorithm is described in detail in Sect. 2.7.2.

There are 3 methods in the study. (1) Water retention measurement is a laboratory experiment for the determination of the water retention characteristic of soil, that is, the evolution of volumetric

water content with soil matric potential. (2) Pore network modeling is a method that uses the pore network representation of pore space obtained through  $\mu$ CT imaging and that can be used to simulate water retention and thus to assess the success and performance of the pore network generation process. (3) Parameterization of the water retention characteristic is needed, for example, for the description of soil water retention in biogeochemical models, and the parameters of the van Genuchten model can be used to characterize the water retention properties of soil. The model also presents a functional form of the water retention characteristic.

All the measured volumetric water content and air-filled porosity values presented in this study were obtained with Eqs. (1) and (2), and all the simulation results for the evolution of air-filled porosity with matric potential were obtained with the drainage percolation algorithm of the OpenPNM package. The van Genuchten parameters are only presented, for reference, in Table 1.

We have added an introductory sentence in Sect.2.7.2.

L186–187: "We used pore network modeling to assess the performance of the pore network approach against the measured water retention properties and to perform further simulations in the peat pore networks."

Line 203: does the "slight vertical shrinkage" here refer to shrinkage during imaging, or some other stage of the experiment? Were the samples imaged after each pressure that was applied to them?

The shrinkage refers to the shrinkage with decreasing volumetric water content during the water retention experiment. To clarify the issue, we have mentioned in Sect. 2.3, which describes the imaging procedure, that the imaging was performed after the retention experiment.

L131–132: "After the water retention experiment, the soil samples [...] The samples were at -10 kPa matric potential during the imaging."

Line 214: delete "better"

Corrected.

Line 234: Please include the text below in the same paragraph as the text above. the topic doesn't really change here.

Corrected.

Line 258: Please indicate which equation was used for calculating the results presented here.

The equations (1) and (2) for volumetric water content and total porosity, respectively, were used, and the air-filled porosity was calculated by subtracting the volumetric water content from the total porosity. We have clarified the sentence.

Line 259-261: Why is it important to state the percentages here?

The shrinkage percentages illustrate the size and shape of the samples during imaging, at -10 kPa matric potential. As it is seen later in the results and discussion, peat shrinkage had a large effect on the performance of  $\mu$ CT imaging and on the performance of the pore network extraction and pore

network simulation methods. Also, the results of the shrinkage measurement described in Sect. 2.2 are presented here.

Line 262-263: This sentence seems like it explains the reason behind the first statement. Please reorganize.

The sentence does not refer to the previous statement but is an introduction for the following sentences.

Line 265: Please add "measured void space" to indicate whether the discussion is about modeled or measured values.

The sentence does not refer to the air-filled volume determined through water retention measurements but to the void space volume calculated from the binary (solid–void) images. We have clarified the sentence.

Line 267-268: The last sentence of the paragraph does not seem to be tied to anything. Please reword or delete.

In the determination of terminology in Sect. 2.7.1, we make a general assumption that there is only one cluster (defined as the pore network) extending through the sample domain. In this section (3.1) we state that this was really the case in this study.

L270–272: "The pore networks were the only connected pore clusters that extended vertically through the network domain in each pore system, and therefore, it was justified to use them in the water retention simulations."

Line 270: Please indicate which method or equation was used for calculating the total porosity here.

There is only one method for the calculation of total porosity of the samples described in the manuscript, Eq. (2). The term "sample porosity" refers to the experimental determination from sample weight and sample volume, "image porosity" or "image void fraction" refers to the fraction of the void voxels in the binary image, and "network porosity" refers to the total volume of the pores in the extracted pore network divided by the network domain volume. All these definitions are found in the text. We have clarified the sentence.

L274: "The total porosity of the peat samples calculated using Eq. (2) differed significantly between sampling depths [...] "

Line 271: does "rather small" refer here to that the difference was too small to have any real-life meaning or consequences?

"Rather small" refers to the fact that the difference was statistically significant but that it was still not very large quantitatively.

Line 280: Please discuss the hysteresis results in more detail. They are mentioned in the abstract and based on that, the reader assumes that they are one of the main results of the study, but here they are mentioned just suddenly in one sentence.

The result in this section, 3.2. are related to the cylindrical network domain and illustrate the hysteresis in these networks, which represent the whole peat samples. Hysteresis also affects the phenomena discussed in Sect. 3.5. We have emphasized in Sect. 3.5 that the hysteresis phenomenon is behind the observed differences in the dynamics of the air-filled volume fraction.

L335–336: "To illustrate the effect of hysteresis, the volume fraction of the connected network was calculated for both imbibition (wetting) and drainage (drying)."

Line 286: Here the term "subsample" that was defined earlier could be actually used.

Corrected.

Line 299: "differed significantly with depth"

Corrected.

Lines 311-315: The results listed here seem very obvious. The text reads as a list of characteristics of peat samples that one would assume to obtain without any context and reference for what one should assume. This all could be fixed by moving the text from discussion about implications of anaerobic pocket formation on methane transport to the introduction and forming some hypotheses for what would be expected based on the peat structure.

In our opinion, the results are not obvious. We found that the dependence of pore size on porosity was weaker in deeper layers and that network connectivity did not depend on porosity in deeper layers as much as it did near the soil surface. The depth dependence of the spatial variation of average pore volume is discussed further in the second to last paragraph of Sect. 4.3 in the Discussion. These results are also related to the aim number 3 of the study, which is to assess the capability of network theory metrics to describe the physical structure of peat pore space.

The text in the discussion is not about the implications of anaerobic pocket formation for methane transport but on the implications of pore structure for anaerobic pocket formation and methane production. These issues are part of the results of this study, and therefore, they cannot be presented in the Introduction.

Line 329-335: This section seems to present the most important results in relation to CH4 production and transport. Please consider moving to the beginning of the results.

The larger, cylindrical sample domain is studied in Sects. 3.1 and 3.2, and the smaller, cubical sample domain (subsample) is studied in Sects. 3.3–3.5. Therefore, it is not suitable to move Sect. 3.5 to the beginning of the Results section. In addition, the Sects. 3.1 and 3.2 illustrate and assess the performance of the pore network simulation method by comparing the simulations to experimental results and thus "validates" the applicability of the simulation method, and the further analysis, which is based solely on simulations, then follows in Sect. 3.5. In addition, this section (Sect. 3.5) does not present any results in relation to CH4 transport as such, as it only discusses the volume available for CH4 production.

Line 347: Please delete "rather" before "well"

Corrected.

Line 366: Please give an estimate how much this darkening affected your results and how it was dealt with. Were the "false" voids removed by hand?

It is very difficult to give a quantitative estimate of the effect of the darkening on the obtained void fractions of the images. This error only concerns the air-filled porosity estimates of the whole samples presented in Fig. 3 because the top and bottom regions were excluded from the pore network domains. The "false voids" were not always individual regions: the darkening also slightly increased the dimensions of existing void space. Therefore, it was not possible to remove the extra void space by hand. We have now emphasized in the text that the darkening only affected the results presented in Fig. 3.

Line 383: Please give an estimate for how much the peat shrinkage affected the simulations. Does shrinkage here refer to the samples and values obtained from them or something that happens during the simulations?

We consider that the paragraph in question describes the effect of peat shrinkage very thoroughly in a qualitative sense. It is very difficult to give a quantitative estimate. Shrinkage refers to the shrinkage of the samples during the water retention experiment in the laboratory before the  $\mu$ CT imaging. We have clarified the sentence.

L388: "Peat shrinkage during the water retention experiment also affected the results of the water retention simulations."

Line 388-394: Please indicate whether the discussion here is about imaged data or the other the measurements.

We have clarified the text.

L393–399: "Thus, a fraction of the pore throats that were air-filled in the samples at -3 kPa matric potential may have shrunk so that they were not detectable in the  $\mu$ CT images constructed at -10 kPa matric potential. This may have generated disconnected pore space in the images and decreased the total volume of the extracted pore network. Also, the shrinkage of the samples may have decreased the dimensions of the pore space so that a higher external pressure was needed for air invasion in the simulations. Conversely, the horizontal shrinkage of some of the samples created continuous void space near the cylinder wall at -10 kPa matric potential, and thus the extracted pore network contained pore space that had presumably not been present in the samples at higher matric potential conditions."

Line 407: Please tell where the result referred to here can be seen, and add a paragraph break after the last sentence on this page.

The sentence ("According to our results...") was meant to refer to the discussion in the following paragraphs. We have added a reference to Fig. 8. We consider that the last sentence of this paragraph ("The distribution and spatial coverage...") gives reasons for the previous sentence and serves as a concluding remark for the paragraph and as an introduction to the remaining subsection. We have now emphasized the connection between these two sentences.

Line 410: Please include the paragraph starting on this line to the paragraph above.

See previous comment.

Lines 420-451: This is a lengthy list of findings and feels not connected to the goal of the manuscript. Please consider explaining most of this in the results, so that in the discussion all this can be directly tied to gas transport like in lines 452-462.

The goal of the manuscript is not only to evaluate the network metrics from the point of view of gas transport but also to assess the capability of complex network theory metrics to describe the physical structure of peat (aim number 3 in the Introduction). Thus, we consider that the issues discussed in L427–458 are directly connected to one of the main goals of the manuscript. In the revision of the manuscript, we have shifted the main aims from gas transport and methane production more towards the description of the physical peat structure by the network metrics as it was what the referees suggested in the first revision round. Because we have introduced the network metrics such as closeness centrality and betweenness centrality, we consider that discussion on the nature and applicability of these metrics is justified. Also, we discuss the reasons behind our finding that local and global connectivity metrics do not always properly describe the structure and properties of a peat pore network, which is stated in L412–416.

Line 464: "estimate the diffusion" sounds a bit strong here because no real quantitative estimate is provided. Maybe use word "imply"?

We have changed the word 'estimate' to 'characterize'.

Line 473: Do you mean "does not promote and does not restrain" or " does not promote nut restrains"? The first does not really make any sense.

We meant that the orientation of the diffusion paths is such that gas transfer (1) is not faster in the vertical direction than in the horizontal direction and (2) is not slower in the vertical direction than in the horizontal direction. The main idea was that the orientation of diffusion is isotropic and there is no significant difference in gas transfer rates between different directions. We have clarified the sentence.

L479–480: "According to our results, the orientation of diffusion paths in pore networks is such that it does not restrain gas transfer towards the atmosphere in deeper, more degraded peat layers."